# Multilevel logistic regression modelling to quantify variation in malaria prevalence in Ethiopia

**Bereket Tessema Zewude**👁️[ID]℗¤*, **Legesse Kassa Debusho**℗, **Tadele Akeba Diriba**℗

Department of Statistics, University of South Africa, Johannesburg, South Africa

👁️ These authors contributed equally to this work.
¤ Current address: Department of Statistics, College of Natural and Computational Science, Wolaita Sodo University, Wolaita Sodo, Ethiopia
* bereket.tessema2010@gmail.com

## Abstract

### Background

Ethiopia has low malaria prevalence compared to most other malaria-endemic countries in Africa. However, malaria is still a major public health problem in the country. The binary logistic regression model has been widely used to analyse malaria indicator survey (MIS) data. However, most MIS have a hierarchical structure which may result in dependent data. Since this model assumes that conditional on the covariates the malaria statuses of individuals are independent, it ignores potential intra-cluster correlation among observations within a cluster and may generate biased analysis results and conclusions. Therefore, the aim of this study was to quantify the variation in the prevalence of malaria between sample enumeration areas (SEAs) or clusters, the effects of cluster characteristics on the prevalence of malaria using the intra-class correlation coefficient as well as to identify significant factors that affect the prevalence of malaria using the multilevel logistic regression modelling in three major regions of Ethiopia, namely Amhara, Oromia and Southern Nations, Nationalities and Peoples' (SNNP).

### Methods

Dataset for three regional states extracted from the 2011 Ethiopian National Malaria Indicator Surveys (EMIS) national representative samples was used in this study. It contains 9272 sample individuals selected from these regions. Various multilevel models with random sample SEA effects were applied taking into account the survey design weights. These weights are scaled to address unequal probabilities of selection within clusters. The spatial clustering of malaria prevalence was assessed applying Getis-Ord statistic to best linear unbiased prediction values of model random effects.

### Results

About 53.82 and 28.72 per cents of the sampled households in the study regions had no mosquito net and sprayed at least once within the last 12 months, respectively. The results

**Data Availability Statement:** The data that support the findings of this study are available from Ethiopian Public Health Institute, but restrictions apply to the availability of these data, which were used under license for the current study, and so

are not publicly available. Data are however available from the authors upon reasonable request and with permission of Ethiopian Public Health Institute (https://ephi.gov.et/).

**Funding:** The author(s) received no specific funding for this work. In addition, the first author acknowledges the financial support from the University of South Africa in the form of Doctoral Bursary.

**Competing interests:** The author(s) have declared that no competing interests exist.

of this study indicate that age, gender, household had mosquito nets, the dwelling has windows, source of drinking water, the two SEA-level variables, i.e. region and median altitude, were significantly related to the prevalence of malaria. After adjusting for these seven variables, about 45% of the residual variation in the prevalence of malaria in the study regions was due to systematic differences between SEAs, while the remaining 55% was due to unmeasured differences between persons or households. The estimated MOR, i.e. the unexplained SEA heterogeneity, was 4.784. This result suggests that there is high variation between SEAs in the prevalence of malaria. In addition, the 80% interval odds ratios (IORs) related to SEA-level variables contain one suggesting that the SEA variability is large in comparison with the effect of each of the variable.

## Conclusions

The multilevel logistic regression with random effects model used in this paper identified five individual / household and two SEA-level risk factors of malaria infection. Therefore, the public health policy makers should pay attentions to those significant factors, such as improving the availability of pure drinking water. Further, the findings of spatial clustering provide information to health policymakers to plan geographically targeted interventions to control malaria transmission.

## Introduction

The estimated malaria cases in Ethiopia has declined from about 7.7 to 2.4 millions and estimated deaths due to malaria has declined from 14,424 to 4,757 between 2010 and 2018. The country has low malaria prevalence compared to most other malaria-endemic countries in Africa (e.g. Burkina Faso, Cameroon, the Democratic Republic of the Congo and Ghana) [1]. However, malaria is still a major public health problem in Ethiopia [2].

The estimated deaths due to malaria also globally declined from 585 000 to 405 000 cases between 2010 and 2018. According to the latest World Health Organization (WHO) Report 2019, the highest reduction in malaria deaths occurred in African region, which was from 533 000 cases in 2010 to 380 000 cases in 2018 [1]. However, according to the same report, the rate of reduction of malaria mortality was slower in the period 2016–2018 than in the period 2010–2015. In year 2018, there were an estimated 228 million cases of malaria occurred worldwide, compared with 251 million cases in 2010 and 231 million cases in 2017 [1].

In most malaria indicator surveys (MIS), the survey designs organize the study populations into clusters such as districts, villages or enumeration areas. Then select households within the clusters to collect data (e.g. see [3, 4]). The data generated from these surveys have a hierarchical or multilevel structure, where individuals in sampled households are nested within enumeration area or cluster and the surveys result in dependent data. Since households that live in the same clusters share similar environment and the public health service facilities in the area, they are likely to resemble each other with respect to the effects from these. Thus, individuals from two randomly selected households from the same cluster or enumeration area, they may have similar malaria status (tested positive or negative) than the malaria status of individuals from two randomly selected households from different clusters, even when the measured household or individual-specific characteristics of the selected households are identical to one another.

Traditionally, the standard binary logistic regression model has been widely used to analyse malaria status data. However, because this model assumes that conditional on the covariates the malaria status of individuals are independent, it disregard potential intra-cluster correlation among observations within a cluster and therefore may generate biased results and conclusions [5, 6]. This model also prevents to study the association between specific cluster characteristics and the individual outcome [7] and ignores the hierarchical structure of the data. The intra-cluster correlation can serve as a meaningful information that can have implications for health care policy makers. Multilevel models incorporate cluster-specific random effects that account for the dependency of the observations by partitioning the total individual variance into variation due to the clusters and the individual-level variation that remains [8–11]. Since the multilevel models take into account the clustered nature of the data, they can correctly estimate standard errors and lead to more accurate inferential decisions and allow to investigate sources of variations within and across clusters.

Multilevel logistic regression models allow analysts to take into account the clustered nature of the data, to investigate sources of variations within- and between-clusters, to describe which variables predict individual differences and to describe which variables predict cluster-level differences. There are two multilevel studies in Ethiopia, Peterson et al. [12], included village as a cluster and concluded that village-level effects were important and Woyessa and Deressa [13] considered household as a cluster and investigated individuals and household effects but their study was conducted within one city in Oromiya region. The studies in Peru [14] and Cambodia [15] also included household as a cluster in their multilevel analyses. The later showed the importance of household-level risk factors in explaining differences in infection prevalence between households and between villages. Whereas, the former compared risk factors in community-specific models and reported malaria prevalence differences between communities were influenced by individual and household variables but variances of the random household-level effects were too small. Recently [16, 17] have also applied the multilevel modelling to malaria survey data, however, the former used the sample weight to compute descriptive statistics not for modelling and the later also fitted unweighted multilevel models. Although there are various studies reported in literatures using MIS data, the multilevel logistic regression models are rarely applied (e.g. see [18–24]). Some of these studies apply either the generalized linear mixed model by introducing random effects to take into account the within-cluster correlation only (e.g. see [18, 20]) or the generalized estimating equation (GEE) (e.g. see [19, 23, 24]), by considering the intra-cluster correlation in order to improve estimates of the fixed effects and their standard errors. Rodriguez and Goldman [25], however emphasized that the estimation of similarity of observations within a cluster or group is not only improve these two estimates but also for yielding important substantive information, in particular by introducing individual level or cluster-level factors as a set of control variables, the analysis may provide information on how the cluster level factors influence the individual characteristics or vice versa. Unlike multilevel models, GEE [26, 27] had been developed to handle correlated data without explicitly accounting for heterogeneity across clusters and it does not provide direct estimates of the variance structure, but treat these as nuisance parameters [28]. Also, it does not allow analysts to examine the influence of individual-level or cluster-level factors on between cluster variation, and sources of intra-cluster correlation [29].

The main objective of the current study is to quantify the variance component of survey enumeration areas (SEAs) or cluster effects and SEAs characteristics on individual malaria rapid diagnostic test (RDT) outcome. In addition to quantifying the variation, we are also interested to identify significant factors that affect the malaria RDT outcomes of household members or the prevalence of malaria in three major regions of Ethiopia. The methods used in this study will allow to make comparisons of the prevalence of malaria between different SEAs

whose characteristics differ from one another and to quantify the heterogeneity in these characteristics. In this study, survey enumeration area (SEA) was considered as a cluster and SEA and cluster, and the malaria RDT outcomes and the prevalence of malaria used interchangeably throughout the paper.

The rest of the paper is organized as follows. The data and the statistical methods used for analyses of these data are introduced in "Materials and methods" section. The results from applying these methods on the study data are discussed in "Results" section. Finally discussion, and conclusions and pointers for future study are given in "Discussion" and "Conclusion" sections, respectively.

## Materials and methods

### Study data

The data used in this study were obtained from the 2011 Ethiopia National Malaria Indicator Survey (EMIS). The survey was conducted by the Ethiopian Health and Nutrition Institutes and its partners, the Ethiopian Ministry of Health in collaboration with the Central Statistics Agency (CSA), US President's Malaria Initiative (PMI), United Nations Children's Fund (UNICEF), Malaria Control and Evaluation Partnership in Africa (MACEPA/PATH), Malaria Consortium, The Carter Center (TCC), World Health Organization (WHO), and International Center for AIDS Care and Treatment Programs (ICAP).

The EMIS was a large nationally representative survey designed to cover key malaria control interventions, treatment-seeking behavior, malaria prevalence; and also to assess anemia prevalence in children under 5 years of age, malaria knowledge among women, and indicators of socioeconomic status [3]. The survey consisted of a two-stage sample design. The first stage involved selecting clusters from a list of SEAs covered in the 2007 Population Census, these areas made up the primary sampling units (PSUs). Data from three major regional states namely Amhara, Oromia and Southern Nations, Nationalities and Peoples' (SNNP) were used in this study. A total of 335 SEAs where 93 from Amhara, 154 from Oromia and 88 from SNNP regions were included in the analyses. The choice of these three regions was driven by the data-sharing limitations imposed by the Ethiopian Public Health Institute, the owner of the data. However, these regions had covered 77.5% of the national enumeration areas analyzed for the survey report. During the survey, blood samples were taken from all children under five years of age in every sampled household and from persons of all ages in every fourth household per WHO guidelines. The blood sample of a child was taken after obtaining consent from residents and assistance of the parent/guardian of the child. Then malaria parasite testing was done using CareStart™ RDT.

### Response and explanatory variables

The dependent or response variable in this study was an indicator of whether an individual had malaria infection (i.e. tested positive for malaria), where 1 signifies tested positive and 0 for tested negative. The covariates or explanatory variables used to explain the malaria RDT outcomes are defined at two levels, as an individual (and household) characteristics, and SEA or cluster characteristics. As individual level characteristics, we used age and gender with a household background variables. The selected household background variables were based on previous studies (e.g. see [18, 20–22, 30, 31]) and included number of household members, whether household had mosquito nets that can be used while sleeping or not which was a dichotomous variable with the categories yes or no; the number of months since the household sprayed interior walls of the dwelling against mosquitoes; number of sleeping rooms in the dwelling, the dwelling has windows which was a dichotomous variable with categories yes and

no; main source of drinking water which was a three-category variable with the categories unprotected, protected source and piped water; main material of the house wall which was a three-category variable with the categories other type, wood and finished wall (e.g. bricks); main material of house floor which was a three-category variable with the categories earth plastered by dung, rudimentary and finished floor (e.g. cemented); main material of the house roof which was a three-category variable with the categories natural (e.g. thatch / leaf), corrugated iron and wood; household toilet facility which was a three-category variable with the categories no facility, pit latrine and flush toilet; and household wealth index which was a five-category variable with the categories poorest, second, middle, fourth and richest. A SEA level characteristics are region which was a three-category variable with the categories Amhara, Oromiya and SNNP regional states; and median altitude in meters. The SEA was used as a cluster, i.e. level 2, variable and the individual and household variables as level-1 characteristics. The malaria indicator survey data also include sampling weights for survey design at level-1.

The median altitude of a SEA may be a marker for cluster-level factors potentially related to malaria infection, such as climate or environmental conditions, and these factors may affect everyone in the SEA. Note that except for very few SEAs, the SEAs of the above three regions in the Ethiopian National Malaria Indicator Survey overlap with "Kebele" which is the smallest administrative unit of Ethiopia, similar to a neighbourhood or a localized and delimited group of people. As it was discussed by Duncan, Jones and Moon [32], the notions of compositional and contextual effects apply not only when the focus is upon context as geographical setting but also it can be applied when context is seen in terms of administrative setting. In Ethiopia, the public service delivery is under the jurisdiction of the regional states. The regional health bureaus are responsible for administration of public health while the districts are responsible for planning and implementation of services. So, public health administrative processes that had been taken by district or regional authorities to control malaria transmission in the SEAs might affect the positive malaria RDT outcomes of individuals living within a particular SEA. Therefore, in this study we defined the individual and household characteristics or covariates as composition effects whereas the SEA (or cluster) level characteristics, i.e. median altitude of a SEA and region where a SEA belongs as contextual effects.

## Statistical methods

**Multilevel logistic regression model with random effects.** Malaria prevalence is likely to vary in different geographical location either due to environment or lack of public health facilities and these effects enable us to include these unknown variations in the model using random effects. Furthermore, individuals living in the same SEA may be more similar to each other than individuals living in other SEA as they share similar environment, public health facilities in the area and other area characteristics that may condition similar malaria RDT outcome. In addition, there is correlation among malaria status of individuals living in the same households. These introduce intra-class correlation, which is a measure of the degree of similarity among malaria status of members of the same cluster, i.e. SEA or household. However, in the dataset used in this study only one member per household was tested for malaria from 47.12% households in the sample. This could introduce large number of missing cases if a within household variation considered. Therefore, this study employed the multilevel logistic regression model with SEA specific random effects to account for the intra-class correlation and hence quantify the variation in a malaria RDT outcome that is accounted for by the SEA variances.

Let $y_{ij}$ denote the malaria outcome of the $i$th individual in the $j$th SEA or cluster identified by the CareStart™ rapid diagnostic tests (RDT) with probability $\pi_{ij}$, where $y_{ij} = 1$ denotes the individual tested positive, while $y_{ij} = 0$ denotes the individual tested negative for malaria. A multilevel logistic regression model with random effects for the outcome $y_{ij}$ is given by

$$\eta_{ij} = g(\mu_{ij}) = \mathbf{X}'_{ij}\boldsymbol{\beta} + \mathbf{Z}'_j\boldsymbol{\alpha} + b_j, \quad i = 1, \ldots, n_j; \ j = 1, \ldots, m, \tag{1}$$

where $g(\cdot)$ is the link function, $\mathbf{X}_{ij} = (1, x_{1ij}, \ldots, x_{pij})$ is vector of $p$ explanatory variables or covariates measured on the $i$ individual and $\mathbf{Z}_j = (z_{1j}, \ldots, z_{qj})$ is vector of $q$ covariates measured on the $j$ SEA (cluster), $\boldsymbol{\beta}$ and $\boldsymbol{\alpha}$ are vector of fixed regression coefficients or parameters and $b_j$ is a random effect varying over SEAs. It is assumed that $b_j$ is independently and normally distributed with mean zero and variance $\sigma_b^2$, in short $b_j \sim N(0, \sigma_b^2)$. The conditional expectation $\mu_{ij} = E(y_{ij}|\mathbf{X}_{ij}, \mathbf{Z}_j, b_j)$ is linked to the linear predictor $\eta_{ij}$ via a link function $g(\cdot)$ and the conditional distribution of $y_{ij}$ belongs to the exponential family.

**Sampling weights.** The MIS data used in this study include a single overall level-1 weighting variable that incorporates level-2 design issues and the weights account for unequal probability of selection given different population sizes within enumeration areas. The multilevel model can be extended to accommodate weights at different levels (e.g. see [33]) using a pseudo-maximum-likelihood approach. Such an approach is very useful in analyzing survey data that arise from multistage sampling. In addition, in these sampling designs, survey weights are often constructed to account for unequal sampling probabilities, non-response adjustments, and post-stratification. When the sampling weights associated with level-1 units, they could lead to bias in variance components estimators if they are large [34]. Therefore, to correct the weights and reduce bias they were scaled using the two methods, called method 1 and method 2 (e.g. see [33, 35]). Let $n_j^{(1)}$ denote the number of level-1 units in the level-2 unit $j$ and let $w_{i|j}$ denote the weight of the $i$th level-1 unit in level-2 unit $j$. Method 1 scales the weights so that the new weights sum to the effective cluster size [33, 35],

$$\sum_{i=1}^{n_j^{(1)}} \lambda w_{i|j} = \frac{\left(\sum_{i=1}^{n_j^{(1)}} w_{i|j}\right)^2}{\sum_{i=1}^{n_j^{(1)}} w_{i|j}^2},$$

where $\lambda$ is the scale factor and it becomes

$$\lambda = \frac{\sum_{i=1}^{n_j^{(1)}} w_{i|j}}{\sum_{i=1}^{n_j^{(n)}} w_{i|j}^2}.$$

Whereas Method 2 scales the weights so that the new weights sum to the cluster sample size [33, 35]. So that the scale factor is

$$\lambda = \frac{n_j^{(1)}}{\sum_{i=1}^{n_j^{(1)}} w_{i|j}}.$$

As the study data do not have specific weight for level-2, here we did not weight level-2 in the analyses. However, as Rabe-Hesketh and Skrondal [33] indicated scaling level-2 weights has little practical effect. The weighted analyses results across the fixed and variances of the random effects from the two scaling methods were almost identical at two decimal places. Even though the results obtained slightly different between these methods, we had the same

inferential decisions for household- and cluster-level characteristics in each of the fitted model, therefore for brevity we reported here the results from scaling method 1.

The following three weighted multilevel logistic regression models with SEA-specific random effects were fitted to the MIS data. The first was the null model, denoted by $M_0$, which had only cluster-specific random effects $b_j$ and did not contain any individual / household or SEA characteristics, that is

$$logit(\pi_{ij}) = \beta_0 + b_j,$$

where $\beta_0$ is identical for all the SEAs and $b_j$ quantify differences between what is measured on average in the study area and what is measured in each SEA. It was fitted to verify if there is indeed variation between SEAs in individuals malaria RDT outcomes. The second model ncorporated included the individual / household-level explanatory variables, called $M_1$, i.e. it has a form

$$logit(\pi_{ij}) = \sum_{k=0}^{p} \beta_k x_{kij} + b_j,$$

where the coefficients $\beta_k$, $k = 1, \ldots, p$, where $p$ is the total number of coefficients which depends on number of categories of predictors in the model, are fixed effect parameters, $x_{0ij} = 1$ and $b_j$ is as defined in model $M_0$. The third model defined using both individual / household-level and two enumeration area / cluster-specific predictor variables, i.e. it was $M_1$ with two enumeration area / cluster-specific predictor variables (region and median altitude of SEA), called $M_2$ and has a form

$$logit(\pi_{ij}) = \sum_{k=0}^{p} \beta_k x_{kij} + \sum_{l=1}^{q} \alpha_l z_{ljk} + b_j,$$

where $\alpha_l$, $l = 1, \ldots, q$, where $q$ is the total number of coefficients for SEAs level predictors and it depends on number of categories of predictors, are the fixed effects for SEAs-level predictors and $z_{ljk}$ is the $k$th observed value of the $l$th predictor at the $j$th SEA. Note that the regression coefficients or fixed effects $\beta_k$ and $\alpha_l$ in the above models represent the study area average effects whereas the SEA-level variance $\sigma_b^2$ provides an estimate of what could be explained by each SEA-level.

**Population-average regression coefficients and odds ratios.** The interpretations of the fixed effects parameters in models $M_1$ and $M_2$ can be done via odds ratio using the population-averaged interpretation and the subject-specific interpretation. The odds ratios from these models give SEA or cluster-specific measures of associations which are adjusted for the unobserved SEA effect $b_j$. Their interpretations natural for individual / household level covariates, however since there is no within SEA variation in the given SEA characteristic in model $M_2$, i.e. region and median altitude of SEA, these odds ratios are not clear for SEA-level variables [11]. The population-average odds ratio, which can be produced from the generalized estimation equations coefficient estimate, is an alternative method to compare risk of malaria infection between two persons from two different SEAs who are identical in other characteristics apart from the covariate of interest [36]. Under the normality assumption of the random effects $b_j$, the population-average regression coefficients, denoted by $\hat{\alpha}_{(PA)l}$ for SEA

characteristic in model $M_2$, can be approximated from the multilevel analyses results of $M_2$
[37] as

$$\hat{\alpha}_{(PA)l} = \frac{\hat{\alpha}_l}{\sqrt{1 + \left(16^2 \times \dfrac{3}{(15\pi)^2}\right) \times \hat{\sigma}_b^2}},$$

where $\hat{\alpha}_l$ and $\hat{\sigma}_b^2$ are the estimated conditional or SEA-specific regression coefficient and the estimated SEA variance in model $M_2$, respectively. Zeger et al. [37] have developed the above formula to equate coefficients between the cluster (i.e. SEA)-specific and population average models under normality assumption of the random effects. Observe from the above formulae

that the SEA-specific regression coefficient is shrink by a factor $1/\sqrt{1 + \left(16^2 \times \frac{3}{(15\pi)^2}\right) \times \hat{\sigma}_b^2}$,

which is called a shrinkage factor [38].

**Interval odds ratio.** Recall that model $M_2$ contains individual-level (level 1) and cluster-level (level 2) covariates. Since for the cluster-level variables, i.e. region and median altitude of SEA, we are comparing two individuals from different clusters, the effects of these variables have different random effects. Due to this, the usual odds ratio interpretations conditioning on all the covariates in the model and random effect $b_j$ are inappropriate for the cluster-level variables as their corresponding odds ratio are random variables. Therefore, in this study the authors applied the interval odds ratio (IOR) to unify the interpretations of fixed and random effects in a subject-specific approach [11, 38, 39] for the cluster-level independent variables. The IOR is briefly discussed in the following paragraphs, for mathematical derivations of the formula see [38].

The IOR is a measure for quantifying the effect of cluster-level variables when using multi-level logistic regression models [39]. Consider the distribution of the odds ratio comparing two persons whose values of the given SEA level covariate differ by one unit, e.g. median altitude, but who have identical values for the other SEA level covariates and of the individual / household level covariates. The IOR is defined as the interval covering the central 80% of odds ratios between two persons from different SEAs and with different values of SEA-level characteristics. The lower, $IOR_L$ and upper $IOR_U$ bounds of this interval are given by

$$IOR_L = \exp(\alpha_l + \sqrt{2\,\hat{\sigma}_b^2} \times \Phi^{-1}(0.10)$$

and

$$IOR_U = \exp(\alpha_l + \sqrt{2\,\hat{\sigma}_b^2} \times \Phi^{-1}(0.90),$$

where $\alpha_l$, $l = 1, \ldots, q$ is the estimated regression coefficient associated with the $l$th SEA-level covariate, $\hat{\sigma}_b^2$ is the estimated variance of the distribution of the random effects, while $\Phi^{-1}(0.10) = -1.282$ and $\Phi^{-1}(0.90) = 1.282$ denote the 10th and 90th percentiles of a standard normal distribution, respectively. Observe from the above formula that, wide IOR indicates that the variability between enumeration areas, i.e. $\hat{\sigma}_b^2$ is large relative to the effect of the SEA-level covariate. A narrow IOR suggests less variability between SEAs. If the IOR contains one, then the SEA variability is large in comparison with the effect of the SEA-level covariate. If the IOR does not contain one, then the effect of the SEA-level covariate is large in comparison with the unexplained between SEA variation [38].

**Quantifying variation: Variance partition coefficient and median odds ratio.** The odds ratio and the IOR allow to measure the association between the odds of the occurrence of the malaria RDT outcomes and individual / household and SEA-level explanatory variables. They

do not provide a measure of the effect of SEA on malaria outcomes, also called general contextual effects. Therefore, to estimate the SEA effects on malaria RDT outcomes and to quantify the variation in the malaria RDT outcomes between SEAs, we applied the variance partition coefficient (VPC) and the median odds ratio (MOR), respectively.

The effect of the general contextual effects is measured by the VPC or the intra-class correlation coefficient (ICC) [11]. The VPC measures proportion of the total observed individual variation in the outcome that is explained by the between SEA (i.e. cluster) variation. The VPC requires an estimate of the variance at the individual level (level-1) and for the standard logistic distribution this variance is equal to $\frac{\pi^2}{3}$ [10]. Therefore, in the multilevel logistic regression models $M_1$ and $M_2$ with a logit link the VPC or ICC is approximated by

$$VPC = \frac{\hat{\sigma}_b^2}{\hat{\sigma}_b^2 + \frac{\pi^2}{3}},$$

where $\hat{\sigma}_b^2$ is the estimated enumeration area variance. The higher the VPC, the higher is the general contextual (i.e. SEA) effects. For further discussion on VPC see Chapter 14 in [10] or Chapter 4 in [9].

The MOR measures how much variability in the malaria RDT outcomes exists between SEAs by comparing two persons from two randomly chosen, different SEAs [38, 39]. Consider two persons chosen randomly from two different SEAs but with the same values of covariates in the model. The MOR is the median odds ratio between the person at SEA with higher risk of malaria infection and the person at SEA with lower risk of malaria infection. It is a function of the estimated SEA, i.e. cluster, variance $\hat{\sigma}_b^2$ and is given by

$$MOR = \exp(\sqrt{2\,\hat{\sigma}_b^2} \times \Phi^{-1}(0.75)),$$

where $\Phi(\cdot)$ is the cumulative distribution function of the standard normal distribution, $\Phi^{-1}(0.75)$ is the 75th percentile, and $\exp(\cdot)$ is the exponential function [38, 39]. Observe from the above formula that the differences in risk entirely quantified by the cluster-specific random effects and MOR is always greater than or equal to 1, equal to 1 when $\hat{\sigma}_b^2 = 1$. For a theoretical derivation of the formula for MOR see [38].

All the multilevel logistic regression models with random effects were fitted with the PROC GLIMMIX in SAS. Since PROC GLIMMIX in SAS uses the weights provided in the data set for analysis, to use the scaled weights the scaled weights should be provided in the data set [40].

## Results

### Individual, household and cluster characteristics

In this study, there were 9272 individuals who tested for malaria infection. Of these individuals, 2384 (25.74%), 4583 (49.43%) and 2302 (24.83%) of them were from Amhara, Oromiya and SNNP regions, respectively, and of these 47.09 per cent were male (Table 1). The overall mean (SD) age of the individuals, number of household members, number of sleeping rooms, number of months since the household sprayed indoor residual and median altitude were 14.33 (17.25), 5.44 (1.93), 1.31 (0.59), 0.88 (1.97) and 1966.57 (379.75), respectively. In the study regions, about 40.79 per cent of individuals were lived in dwellings that had at least one window. Over half of the sampled households (53.82%) in the study regions had no mosquito net, however about 73.44 per cent of sampled households in Amhara regions had mosquito net. Information about indoor residual spraying (IRS) of interior walls of the dwelling against

**Table 1. Descriptive statistics for variables used in the study by region.**

| Variables | | Region | | | Total |
|---|---|---|---|---|---|
| | | Amhara | Oromiya | SNNP | |
| Number of individuals | N (%) | 2384 (25.74) | 4583 (49.43) | 2302 (24.83) | |
| Age | mean (SD) | 16.148 (18.31) | 13.688 (17.17) | 13.74 (16.10) | 14.33(17.25) |
| Number of household members | mean (SD) | 5.19 (1.88) | 5.47 (1.92) | 5.65 (1.98) | 5.44 (1.93) |
| Number of sleeping rooms | mean (SD) | 1.23 (0.47) | 1.34 (0.63) | 1.34 (0.61) | 1.31 (0.59) |
| Number of months since sprayed | mean (SD) | 0.69 (1.71) | 0.82 (1.93) | 1.18 (2.24) | 0.88 (1.97) |
| Median altitude | mean (SD) | 2091.15 (379.49) | 1916.00 (374.14) | 1938.06 (361.89) | 1966.57 (379.75) |
| Rapid diagnostic test (RDT) result | Positive, N (%) | 65 (2.72) | 56 (1.22) | 105 (4.56) | 226 (2.44) |
| | Negative, N (%) | 2322 (97.28) | 4527 (98.78) | 2197 (95.44) | 9046 (97.56) |
| Gender | Male, N (%) | 1119 (46.88) | 2154 (47.00) | 1093 (47.48) | 4366 (47.09) |
| | Female, N (%) | 1268 (53.12) | 2429 (53.00) | 1209 (52.52) | 4906 (52.91) |
| Availability of windows | No, N (%) | 1367 (57.27) | 2841 (61.99) | 1282 (55.69) | 5490 (59.21) |
| | Yes, N (%) | 1020 (42.73) | 1742 (38.01) | 1020 (44.31) | 3782 (40.79) |
| Household uses mosquito nets | No, N (%) | 302 (26.56) | 1398 (65.36) | 629 (59.85) | 2329 (53.82) |
| | Yes, N (%) | 835 (73.44) | 741 (34.64) | 422 (40.15) | 1998 (46.18) |
| Household used IRS | No, N (%) | 871 (77.08) | 1534 (71.95) | 667 (63.65) | 3072 (46.18) |
| | Yes, N (%) | 259 (22.92) | 598 (28.05) | 381 (36.35) | 1238 (28.72) |
| Main source of drinking water | Unprotected, N (%) | 1262 (52.87) | 2931 (63.95) | 1178 (51.17) | 5371 (57.93) |
| | Protected, N (%) | 560 (23.46) | 693 (15.12) | 482 (20.94) | 1735 (18.71) |
| | Piped water, N (%) | 565 (23.67) | 959 (20.93) | 642 (27.89) | 2166 (23.36) |
| Main material of house wall | Other, N (%) | 6 (0.25) | 338 (7.38) | 274 (11.90) | 618 (6.67) |
| | Wood wall, N (%) | 2100 (87.98) | 3774 (82.35) | 1607 (69.81) | 7481 (80.68) |
| | Finished wall, N (%) | 281 (11.77) | 471 (10.28) | 421 (18.29) | 1173 (12.65) |
| Main material of house roof | Natural roof, N (%) | 809 (33.89) | 2128 (46.43) | 1056 (45.87) | 3993 (43.07) |
| | Wood material, N (%) | 263 (11.02) | 775 (16.91) | 715 (31.06) | 1753 (18.91) |
| | Corrugated iron & Other, N (%) | 1315 (55.09) | 1680 (36.66) | 531 (23.07) | 3526 (38.03) |
| Main material of house floor | Plastered by dung, N (%) | 2228 (93.34) | 4339 (94.68) | 1994 (86.62) | 8561 (92.33) |
| | Rudimentary floor, N (%) | 146 (6.12) | 168 (3.67) | 206 (8.95) | 520 (5.61) |
| | Finished floor, N (%) | 13 (0.54) | 76 (1.66) | 102 (4.43) | 191 (2.06) |
| Type of toilet facility | No facility, N (%) | 853 (35.74) | 1933 (42.18) | 214 (9.30) | 3000 (32.36) |
| | Pit latrine, N (%) | 954 (39.97) | 2263 (49.38) | 1360 (559.08) | 4577 (49.36) |
| | Flush & hanging toilets, N (%) | 580 (24.30) | 387 (8.44) | 728 (31.62) | 1695 (18.28) |
| Household wealth status | Poorest, N (%) | 357 (14.96) | 1276 (13.76) | 247 (10.73) | 1880 (20.28) |
| | Second, N (%) | 514 (21.53) | 956 (20.86) | 546 (23.72) | 2016 (21.74) |
| | Middle, N (%) | 425 (17.80) | 745 (16.26) | 587 (25.50) | 1757 (18.95) |
| | Fourth, N (%) | 595 (24.93) | 853 (18.61) | 393 (17.07) | 1841 (19.86) |
| | Richest, N (%) | 496 (20.78) | 753 (16.43) | 529 (22.98) | 1778 (19.18) |

mosquitoes within the last 12 months prior to the survey was also collected. Only 28.72 per cent of the sampled households had sprayed at least once within the last 12 months. About 57.93 per cent of the sampled household had unprotected source of drinking water whereas 18.71 and 23.36 per cents of them had protected and pipped sources of drinking water, respectively. Almost 80.68 per cent of sampled households had either bamboo / wood with mud or stone with mud as main material of their dwellings wall, 12.65 per cent of the households had finished wall and 6.67 per cent of households' dwellings either had no wall or uncovered abode or bamboo / reed or carton as main material of their dwellings wall. The finished wall group

included cement walls, walls made with stones and cement and bricks. Large per cent (43.07) of the sampled households' dwellings had natural roof, that is either thatch / leaf or rustic mat / plastic sheet as their main roof material, 38.03 per cent of them had either corrugated iron or calamine / cement fiber or cement / concrete and 18.91 per cent of the households' dwellings had either sticks and mud or reed / bamboo or wood planks or wood or roofing shingles as their main roof material. The majority (92.33%) of the households' dwellings had earth / sand or dung plastered floor, 5.61 per cent of the households' dwellings had either wood planks or palm / bamboo as their main floor material and only 2.06 per cent of the households's dwellings had a finished floor which were either ceramic tiles or cement or carpet. Of the sampled households, 49.36 and 18.28 per cents of them had pit latrine and flush toilet facilities in their dwellings, respectively, whereas 32.36 of the households had no toilet facility. The wealth index used in this survey serves as an indicator of level of wealth of a household that is consistent with expenditure and income measures [3]. Of the sampled households 20.28, 21.74, 18.95, 19.86 and 19.18 per cents of them were in the poorest, second poorest, middle, fourth and the richest (or highest) wealth quintile, respectively (Table 1).

## Fitted multilevel models

Before fitting the models $M_1$ and $M_2$, we have used the stepwise variable selection technique to determine the covariates to be included in these models. Then we have checked for the presence of multicollinearity among the selected covariates using the variance inflation factors (VIF), excluding those with a VIF greater than 5. Then the covariates for the final $M_1$ and $M_2$ models were selected by eliminating those implausible applying the likelihood ratio tests, the Akaike Information Criteria (AIC) and the Bayesian Information Criteria (BIC). Accordingly, of the thirteen individual / household characteristics covariates considered, only seven (age, gender, household had mosquito nets, the number of months since the household sprayed interior walls of the dwelling against mosquitoes, the dwelling has windows, main source of drinking water, household toilet facility) were selected for $M_1$, and the two SEA level characteristics (region and median altitude) and the seven covariates of model $M_1$ were selected for $M_2$. The model selection process is provided in the Tables A and B in S1 File. We have also fitted standard logistic regression models ignoring the cluster information or multilevel structure entirely using $M_1$ and $M_2$ variables (see Table C in S1 File). These models were fitted for comparison purpose, otherwise they are completely not suitable for the data in this study. The fit statistics indicate a substantial improvement in the multilevel version of $M_1$ and $M_2$ compared to the logistic regression models (see Table 2 and Table C in S1 File), implying the multilevel models were preferred over the logistic regression models for the study data. Note also that, as discussed in the Background section, the logistic regression resulted in incorrect inference results, for example the coefficient for number of months since sprayed is statistically significant in the logistic regression models but non-significant in the multilevel models.

The asymptotic chi-square mixture distribution [41] test statistic for testing $H_0 : \sigma_b^2 = 0$ against $H_1 : \sigma_b^2 > 0$ in models $M_0$, $M_1$ and $M_2$ takes the values 339.01, 242.19 and 168.87,

**Table 2. Fit statistics for models with cluster-specific random effects $b_j$ only ($M_0$), with the individual / household-level predictors and $b_j$ ($M_1$), and $M_1$ with cluster-specific predictors ($M_2$).**

| Fit statistics | Fitted models | | |
|---|---|---|---|
| | $M_0$ | $M_1$ | $M_2$ |
| -2 log(Lik) | 1703.19 | 1661.00 | 1617.74 |
| AIC (smaller is better) | 1707.19 | 1683.00 | 1645.74 |
| BIC (smaller is better) | 1714.81 | 1724.96 | 1699.14 |

respectively with $p$-value <0.0001 in all the three tests. The large value of the test statistic or a very small $p$-value strongly suggests a rejection of the null hypothesis $H_0 : \sigma_b^2 = 0$ that no cluster-specific random effects should be included in the model. Therefore, these results imply the need for the SEA (cluster)-random effects in the model.

The estimated intercept for model $M_0$ was −5.468 (with a standard error = 0.255), while the estimated variance of the SEA-specific random effects was 4.041 with a standard error of 0.695. Thus, the probability for a person who lived in any SEA of the three regions where the study conducted tested positive for malaria was 0.42% given that the random effect of the SEA was equal to zero on the logit scale. Thus, taking the inverse logit transformation of the interval for $\beta_0$, for 95% SEAs, the SEA-specific probability of a person who lived in the areas had tested positive for malaria would lie in the interval (0.003, 0.007). Note that because some of $b_j$ may not be equal to zero, the average SEA predicted probability of a person is tested positive for malaria may differ from the average SEA-specific probability of malaria infection.

The fit statistics or model selection criteria for fitted $M_0$, $M_1$ and $M_2$ were displayed in Table 2. The values of these criteria suggest that $M_2$ is a preferred model to fit the study. Therefore, in what follows, we only report results based on $M_2$.

The estimated regression coefficients associated with the individual / household and SEA-level characteristics (i.e. $\hat{\beta}_k$ and $\hat{\alpha}_l$ with corresponding standard errors in brackets), the $p$-values to test $\beta_k = 0$ and $\alpha_l = 0$, and odds ratio with their 95% confidence intervals (CI) in brackets for model $M_2$ are given in Table 3. The estimates of odds ratio (OR) of individual tested

**Table 3. Estimated odds ratios (95% confidence intervals) for model $M_2$.**

| Independent variables | Estimate (SE) | $\chi^2$ $p$-value | Odds ratio (95% CI) |
|---|---|---|---|
| Age | -0.014 (0.006) | 0.0254 | 0.986 (0.975, 0.997) |
| Gender, Female | -0.328 (0.149) | 0.0278 | 0.720 (0.538, 0.965) |
| Windows exist, Yes | 0.687 (0.197) | 0.0005 | 1.988 (1.351, 2.925) |
| Number of months since sprayed | 0.019 (0.036) | 0.6060 | 1.019 (0.950, 1.093) |
| Mosquito nets, Yes | 0.444 (0.246) | 0.0710 | 1.559 (0.963, 2.525) |
| Drinking water (Ref Unprotected) | | 0.0389 | |
| Protected source | -0.655 (0.295) | | 0.520 (0.291, 0.927) |
| Piped water | -0.134 (0.236) | | 0.874 (0.551, 1.389) |
| Toilet facility (Ref No facility) | | 0.2284 | |
| Flush toilet | 0.601 (0.298) | | 1.825 (1.018, 3.270) |
| Pit latrine | 0.161 (0.258) | | 1.174 (0.708, 1.947) |
| Region (Ref Amhara) | | <0.001 | |
| Oromiya | -1.045 (0.362) | | 0.352 (0.173, 0.715) |
| SNNP | 0.258 (0.361) | | 1.294 (0.637, 2.627) |
| Median Altitude | -0.002 (0.0004) | < 0.001 | 0.998 (0.997, 0.998) |
| *Variance of random effects* | | | |
| $\sigma_b^2$ | 2.693 (0.643) | | |
| VPC or ICC | 0.450 | | |
| Median odds ratio | 4.784 | | |
| Interval odds ratio | | | |
| Region | | | |
| Oromiya versus Amhara | | (0.018, 6.884) | |
| SNNP versus Amhara | | (0.066, 25.335) | |
| Median Altitude | | (0.051, 19.535) | |

positive for malaria relative to an appropriate reference group was obtained by taking exponential of the estimates (i.e. $\exp(\hat{\beta}_k)$ and $\exp(\hat{\alpha}_l)$).

The *p*-values show that four of the seven individual / household characteristics, namely age, gender, household had mosquito nets, the dwelling has windows and source of drinking water were significantly associated with the odds of malaria infection of individuals living in the study area. Furthermore, the two SEA-level characteristics, region and median altitude were also strongly statistically significant (*p*-value < 0.001).

The results in Table 3 revealed that the odds of malaria decreased as the age of a person increased (OR = 0.986 with 95% CI: 0.975, 0.997). The OR of positive malaria RDT outcome for female person relative to male person was 0.720 with 95% CI (0.538, 0.965). Since these 95% confidence intervals do not contain one, they are suggesting that the odds of positive malaria RDT outcome was significantly decreasing as a person gets older and there was a significant difference between male and female groups. However, female person was less likely infected by malaria in the study area. Individuals who lived in houses with windows were 1.988 times more likely to have had malaria than those lived in houses without windows (95% CI: 1.351, 2.925). As the number of months since the household sprayed interior walls of the dwelling against mosquitoes increased the positive malaria RDT outcome was slightly increased but this increase was statistically nonsignificant (OR = 1.019, 95%: 0.950, 1.093). A person who lived in dwellings with mosquito net was 1.559 times more likely to had malaria than those lived in dwellings without mosquito nets (OR = 1.559, 95% CI: 0.963, 2.525).

Compared to persons who were living in dwellings that had unprotected source of drinking water, the ORs positive malaria RDT outcomes for those in households that had protected source and piped water as the main sources of drinking water were 0.520 (95% CI: 0.291, 0.927) and 0.874 (95% CI: 0.551, 1.389), respectively. The results also show that those individuals residing in households with flush and pit toilets had a higher risk of malaria infection, with ORs 1.825 (95% CI: 1.018, 3.270) and 1.174 (95% CI: 0.708, 1.947), respectively.

**Quantify variation: Variance partition coefficient and median odds ratio.** From Table 3, the estimated between-SEA variance for model $M_2$ was 2.693, which corresponds to VPC of 0.450. This implies that of the residual variation in malaria RDT outcomes of persons that were observed after adjusting for seven individual / household and two enumeration area characteristics, 45.0 per cent was due to systematic differences between SEAs, while the remaining 55.0 per cent was due to unmeasured differences between persons or households. The estimated MOR, i.e. the unexplained SEA heterogeneity, was 4.784. This result suggests that there is high variation between SEAs in malaria RDT outcomes.

To explore the unexplained heterogeneity further and to identify the hot-spots survey enumeration areas or districts in the three regions for targeted interventions, we applied the Getis-Ord $G_i^*$ statistic [42] to best linear unbiased prediction (BLUP) values of the SEAs' random effects (or model random effects). The BLUP values of random effects were obtained after accounting for the covariates. Fig 1 displays the three regions (with other regions) covering the study area (panel a) and map showing hot-spots for the SEAs' random effects (panel b). The plot in panel (b) shows that there were hot-spots in each of the three regions, for example, in Amhara region there were hot-spots in Lay Gayint, Mirab Este, Simada, Machakel, Debre Elias, Anded, Semen Achefe and Chilga districts; in Oromiya region in Dera, Begi, Guto Gida, Sasiga, Merti, Shala, Qercha, Tero Afeta, Dire and Nono districts, whereas in SNNP region in South Ari, Shebedino, Chuko, Misha, Lanifaro and Kindo Koyisha districts.

**Population-average regression coefficients and odds ratios for SEA-level characteristics.** The odds ratios in Table 3 give SEA or cluster-specific measures of associations which are adjusted for the unobserved SEA effect $b_j$. As discussed in the methodology section, their

(A)

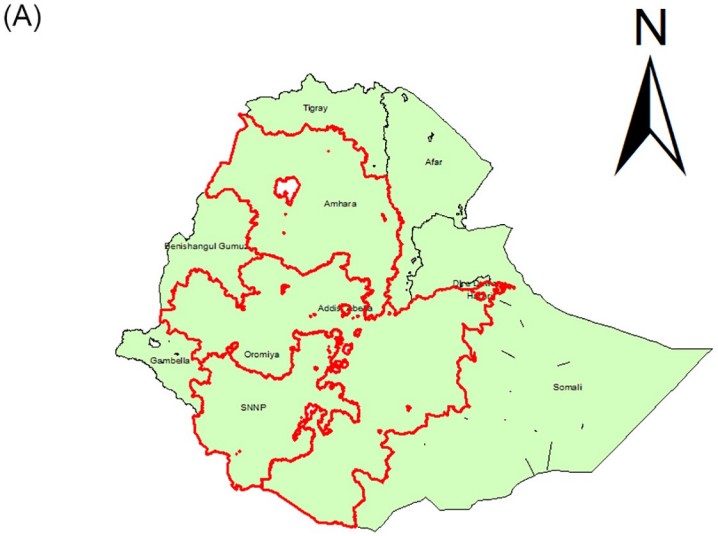

(B)

**Cluster(SEA) random effects**

- ● Cold Spot - 99% Confidence
- ◯ Cold Spot - 95% Confidence
- ● Cold Spot - 90% Confidence
- ◯ Not Significant
- ▲ Hot Spot - 90% Confidence
- ▲ Hot Spot - 95% Confidence
- ▲ Hot Spot - 99% Confidence

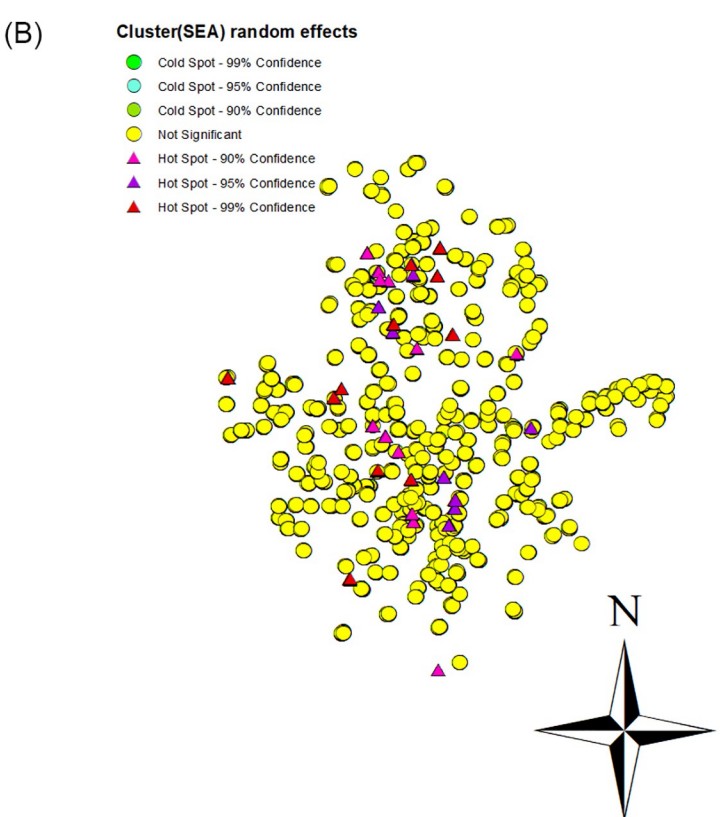

**Fig 1. The three regions covering the study area (panel a) and spatial clustering of the BLUP SEAs' random effects using the Getis-Ord $G_i^*$ statistic.** (a) Ethiopia map, (b) Hot-spots.

interpretations are natural for individual / household level explanatory variables, but these odds ratios are not clear for SEA-level variables. In such case, one can use the population-average odds ratio to compare risk of malaria infection between two persons from two different SEAs who are identical in other characteristics apart from the covariate of interest [36].

Since the estimated variance of the random effects for model $M_2$ is $\hat{\sigma}_b^2 = 2.693$, the corresponding shrinkage factor is 0.720. Since the shrinkage factor is smaller than one, the population-average regression coefficients and their corresponding population-average odds ratios are different from that of the SEA-specific regression coefficients and corresponding odds ratios given in Table 3. In particular, the population-average odds ratios for the SEA characteristics, Oromiya region versus Amhara region, SNNP region versus Amhara region and median altitude of SEA are 0.471, 1.204 and 0.999, respectively. These are different from the SEA-specific odds ratios displayed in Table 3. These odds ratios are for comparisons of persons of the same individual / household-level characteristics living in different enumeration areas.

**Interval odds ratio.**   Recall from Table 3 that the SEA explanatory variables coefficients in model $M_2$ are -1.045, 0.258 and -0.002 and SEA-specific variance is 2.693. Therefore, the 80% IORs for the two SEA-level characteristics (region and median altitude) are (0.018, 6.884) for Oromiya region versus Amhara region, (0.066, 25.335) for SNNP region versus Amhara region and (0.051, 19.535) for median altitude. These results, for example, imply that if one is randomly selecting two persons with identical characteristics, one from SNNP region and one from Amhara region and comparing their odds of having malaria infection, the odds ratios of malaria infection will lie within the interval (0.066, 25.335) in 80 per cent of such comparisons. Since all three intervals contain one, therefore the enumeration area variability is large in comparison with the effect each of the two SEA-level variables. This supports the observation made from the MOR. The intervals also imply that these characteristics do not account for a substantial amount of the SEA heterogeneity. In addition, the intervals, in particular those of SNNP region versus Amhara region and median altitude are wide, indicating that the variability between SEAs is large relative to the effects of region and median altitude. These wide intervals are reflecting a large amount of unexplained variation between SEAs in the malaria RDT outcomes and suggesting other SEA-level variables are needed to explain the SEA heterogeneity.

## Discussion

This paper attempts to provide multilevel logistic regression models with random effect as the best alternative to analyze the malaria indicator survey data compared to statistical techniques that researchers commonly use for correlated data, e.g. the generalized estimating equation (GEE). The GEE does not explicitly account for heterogeneity across SEAs, i.e. between cluster variation or provide direct estimates of the SEAs variance structure rather it consider these as nuisance parameters. Furthermore, unlike the multilevel logistic regression models with random effect with the GEE one cannot assess the effect of individual / household-level or SEA-level factors on the SEA-level variation.

The results of this study clarified the association between the malaria RDT outcomes of individuals and their age and gender, that their vulnerability to malaria infection decreases significantly with increasing age. Females were less likely infected by malaria in the study area. The result on age agrees with the findings of [19, 20, 31, 43–46] and the gender finding is similar to those of Baragetti, et al [21, 46, 47], Roberts and Matthews [19] and Ugwu and Zewotir [20].

The household characteristics, specifically socio-economic factors such as whether the dwelling had windows or not and the household main source of drinking water were significantly positively associated with household member's malaria infection. The positive

association of positive malaria RDT outcomes with the dwelling had windows could be that windows created more entry points for the mosquito if these windows were not closed properly or during peak biting times.

The number of months since the household sprayed interior walls of the dwelling against mosquitoes before the survey was positively associated with an individual's malaria RDT outcome but the association was statistically nonsignificant. The positive association implies that the shorter the period since a household sprayed the lower the risk of malaria infection. Researches show that the indoor residual spray (IRS) associated with reduced malaria risk [48, 49]. The nonsignificant association result in the current study could be due to low per cent of the sampled households (28.72%) had sprayed at least once within the last 12 months (i.e. Household used IRS in Table 1) and in the study data, for individuals from households which had not sprayed indoor residual, the number of months since the household sprayed interior walls before the survey was captured as zero month. However, the variable Household used IRS was not included in the fitted models because it had high correlation with some of the predictor variables. Although, it was non-significant, the positive association of the number of months since the household sprayed interior walls of the dwelling against mosquitoes before the survey with an individual's malaria RDT outcome in the current study supports the findings of the above researches.

The current study showed that there was a positive association between household has mosquito nets and the prevalence of malaria in the study regions, this contradicts results reported in malaria literature. However, this association was statistically nonsignificant. The positive associate of having mosquito nets with the prevalence of malaria could be because mosquito nets are delivered to areas with high malaria burden or due to either households did not treat the nets with insecticide or inappropriate use of the nets or perhaps an individual or a household member was exposed to mosquito bites during other times of the day or evening when the net was not in use. This also could be due to household members did not get proper training on how to use the nets from local public health workers. In addition, less than half of the sampled households in the study regions had no mosquito nets, this may increase the vulnerability of individuals living in the households which had no mosquito nets to malaria infection. Therefore, the regional government should frequently provide IRS to households and mosquito nets to those households which do not have in malaria hot-spots areas and the population at risk, and further educate people about their advantages. The results also showed that good quality source of drinking water was associated with a lower risk of malaria compared to unprotected source. This result similar to the results of [17, 50] showing there was a significant association between unprotected water sources and malaria prevalence. The unprotected sources such as irrigation ditches / channels, or dam can create breeding sites for larvae [51]. This may suggest that the regional governments should expand the provision of protected water sources or piped water to malaria hot-spots areas [17]. There was a significant negative association between positive malaria RDT outcomes and median altitude, which showed that the odds of malaria for individuals in the study regions decreased with an increase in median altitude. This agrees with the studies conducted in Ethiopia [16], Kenya [52], Uganda [19] and Ghana [17].

Beside identifying significant risk factors associated with malaria infection, the analyses allowed us to quantify heterogeneity in odds ratio scale by using characteristics from the odds ratios between pairs of randomly selected persons from different SEAs. The SEA-level variance decreased as the individual / household- and SEA-level characteristics were introduced in the fitted models suggesting that when accounting for the individual / household- and SEA-level characteristics, part of the variability which is relevant at the SEA-level (level-2) becomes lower. This also means that the SEA-level variance quantifies part of the variability which is

relevant at SEA-level but not explained by SEA-level characteristics or factors introduced in the model.

The 80 per cent IORs for the SEA-level characteristics in model $M_2$ contain one implying these characteristics do not account for a substantial amount of the SEA heterogeneity. The intervals are wide, indicating that the SEA variability is large in comparison with the effect each of the two SEA-level explanatory variables. Even though the MOR reduced when the SEA explanatory variables introduced in model $M_2$ compared to other fitted models, but its value was high. Hence, still there is a large amount of unexplained variation between SEAs in the malaria RDT outcomes. Overall this suggests that other SEA-level variables are needed to explain the SEA heterogeneity. Furthermore, the presence of hot-spots in the spatial patterns of SEAs' random effects imply that not all of the SEA-level heterogeneities in malaria prevalence in the study regions were explained by the individual / household and SEA characteristics. The spatially heterogeneity in malaria prevalence indicates an imperative for more regular surveillance to detect emerging and existing malaria hot-spots areas for targeted health interventions by authorities to prevent malaria transmission, especially in settings where there are limited resources.

## Conclusion

In the present study, the multilevel logistic regression with random effects allowed us to describe the effects of the different sources of heterogeneity and associations between malaria RDT outcomes. The models yield odds ratios that have SEA-specific, which have within SEA interpretation. The methods used in this study, such as the estimator of population-average effect of SEA characteristics and IOR, which is the summary measure of the effects of these characteristics have allowed to make comparisons of malaria RDT outcomes between different SEAs whose characteristics differ from one another and to quantify the heterogeneity in these effects. The VPC or ICC and the MOR have allowed us to quantify the magnitude of the effect of SEA or the general contextual effect, by quantifying the magnitude of the variation in malaria RDT outcomes between SEAs. Further, the spatial clustering analysis on the BLUP of model random effects help to verify that there is unexplained heterogeneity in malaria prevalence in the study regions and to identify malaria hot-spots. Such approach yield more extended information that can be helpful in public health policy, for example, estimation of the extent to which individuals within a given SEA are correlated with one another in relation to malaria RDT outcome, i.e. estimation of VPC (or ICC), may provide information about the efficacy of focusing intervention on SEAs instead of individuals. Beside quantifying the variation, the models used allowed us to identify significant risk factors associated with malaria infection. Therefore, the public health policy makers should pay attentions to those significant factors, such as improving the availability of pure drinking water. In addition, they should provide households mosquito nets and IRS, and should train households to spray the interior walls of their dwellings against mosquitoes more frequently. The results also show regional variation in malaria infection prevalence, thus special attention should be given to those people living in districts that were identified as hot-spots.

Malaria presence depends mainly on climatic factors such as temperature, humidity, and rainfall. Malaria is transmitted in tropical and subtropical areas, where *Anopheles* mosquitoes can survive and multiply, and where malaria parasites can complete their growth cycle in the mosquitoes. Human behavior factors, often dictated by social and economic reasons, can influence the risk of malaria for individuals and communities. For example, human activities can create breeding sites for larvae such as standing water in irrigation ditches and burrow pits; and population movement and migration [53]. In this study, however the general contextual

factors were defined at the SEA level where the individual / household resided. Therefore, relative to people activities and mobility, this level could hide part of the environment. As many dimensions and determinants of environment interrelated, interpretation of results concerning area factors is usually complex [54]. The differences between SEAs could be due to bio-ecological or human factors. Further, the climate could affect the malaria cycle and vector development. The inclusion of these factors in the multilevel models may explain some of the SEA heterogeneity, but these were not introduced in the models because the MIS data do not contain these measurements. However, these factors are strongly correlated with altitude, which synthesizes climatic and vegetal conditions. Therefore, part of the altitude influence certainly reflects the influence of climatic factors on malaria. In this paper, our analysis was done using data extracted from EMIS for three regions, therefore the results should be interpreted with caution. The findings are limited to these regions and not necessarily generalizable to the country.

## Supporting information

**S1 File. Multilevel logistic regression modelling to quantify variation in malaria prevalence in Ethiopia.**
(PDF)

## Acknowledgments

We thank the Ethiopian Health and Nutrition Research Institute for giving access to the data. In addition, the first author acknowledges the financial support received from the University of South Africa in the form of Doctoral Bursary.

## Author Contributions

**Conceptualization:** Bereket Tessema Zewude.

**Methodology:** Bereket Tessema Zewude.

**Software:** Bereket Tessema Zewude, Legesse Kassa Debusho.

**Writing – original draft:** Bereket Tessema Zewude.

**Writing – review & editing:** Bereket Tessema Zewude, Legesse Kassa Debusho, Tadele Akeba Diriba.

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
