## [Decision Letter · Decision Letter 0]

14 Jul 2021

PONE-D-21-12914

Multilevel Logistic Regression Modelling to Quantify

Variation in Malaria Prevalence in Ethiopia

PLOS ONE

Dear Dr. Zewude,

Thank you for submitting your manuscript to PLOS ONE. After careful consideration, we feel that it has merit but does not fully meet PLOS ONE’s publication criteria as it currently stands. Therefore, we invite you to submit a revised version of the manuscript that addresses the points raised during the review process.

Please can you address the following comments I have as Academic Editor:

Overall the English language is good but the manuscript needs a final copy-edit before publication. E.g. 2.4 million (not millions) in the abstract. Line 36 should read ‘may have *more* similar’. There are other small errors.

Introduction: Please update the malaria burden numbers from 2018 to more recent data (e.g. using the latest World Malaria Report)

Table 1. Please briefly explain in the legend the difference between the 3 models. The table should be broadly understandable to the reader without referring to the main text.

The abstract is rather hard to follow for a generalist reader. Please consider rewriting this sentence: “The main objective of this study was to quantify the variance component of cluster effects and cluster characteristics on individual malaria rapid diagnostic tests (RDT) outcome.”   Perhaps you could say something like ‘to quantify the extent of clustering of malaria infection as assessed by RDT’.

 Similarly the results section of the abstract is hard to read. Can you name some of the most important variables for explaining the clustering, which you found?

I suggest moving the final paragraph which discusses malaria climate further up in the discussion, since this is a key point. Please reference other studies which have sought to explain malaria prevalence by both climate variables as well as MIS variables. For example the Malaria Atlas Project is a major group which has been working on this problem for many year. E.g. see the map tab on the Malaria Atlas Project forecasts of malaria prevalence for Ethiopia https://malariaatlas.org/trends/country/ETH  What are the differences in the variables used for analysis in this study or other similar geospatial models compared to yours?

I suggest removing the phrase ‘best alternative’ for analysis on line 398, since there are many other sophisticated methods which seek to do this, e.g. geospatial modelling as referenced above.

We look forward to receiving your revised manuscript.

Kind regards,

Lucy C. Okell

Academic Editor

PLOS ONE

Journal Requirements:

Additional Editor Comments:

Reviewers' comments:

Reviewer's Responses to Questions

**Comments to the Author**

1. Is the manuscript technically sound, and do the data support the conclusions?

Reviewer #1: Yes

2. Has the statistical analysis been performed appropriately and rigorously? 

Reviewer #1: Yes

3. Have the authors made all data underlying the findings in their manuscript fully available?

Reviewer #1: Yes

4. Is the manuscript presented in an intelligible fashion and written in standard English?

Reviewer #1: Yes

5. Review Comments to the Author

Reviewer #1: Review for "Multilevel Logistic Regression Modelling to Quantify

Variation in Malaria Prevalence in Ethiopia"

PLOS one

PONE-D-21-12914

Overview

---------

This manuscript analysis a good sized data of malaria prevalence in Ethiopia (MIS data).

The stated main aim of the analysis is to estimate the variances of random effects terms that describe variation between clusters that is unexplained by covariates.

As malaria is of course an important public health problem, analyses that can inform policy makers are always useful.

The analysis is well conducted and care is taken to account for some of the important diffulties present in this type of data.

Overall I think the analysis is a useful contribution to the literature.

I have some minor points that I have detailed below.

I have also included a fairly large number of language edits.

Minor points

--------------

My main problem with the manuscript in it's current state is that the policy importance of the main aim of the analysis is not well explained and is not trivially obvious.

What policy implications does a large (or small) between-cluster variance component have?

I can imagine some.

Maybe it suggests that we are still missing covariates important for predicting malaria.

If so, what might those covariates be and what are the barriers to collecting that data?

Is it that nonlinear models and interaction terms are needed?

Or is it that there are cluster level drivers that we will never be able to measure?

What does this mean for policy, that we an never make excellent predictions of malaria or that the only way to guide resource distribution is to measure malaria prevalence in ever cluster in a country?

Relatedly, while you cite some references that use random effects models, you don't cite any estimates of between-cluster variance from any other study.

While I recognise that these estimates might be difficult to find in many papers, if this is the main aim of the analysis, any previous attempts at estimating this value should be mentioned, if nothing else to help us set our expectations.

Have previous estimates found that between-cluster variation is very high or very low?

I'm not sure if this point is relevant but in some senses, the cross-validation error in models of malaria prevalence are a measure of the same quantity.

It is the amount of malaria prevalence that cannot be predicted with the covariates in the model.

The paragraph in lines 80-84 seems very unecessary to me. The structure of the paper is totally standard.

Lines 243-245. I'm not sure what this line means. Are you saying that you needed to seperately calculate the weights and tell PROC GLIMMIX to use them? Or that you have provided the weights in an attached dataset? Or something else.

The section in lines 251-257 that explain how variable selection was performed should be in the methods somewhere.

Lines 367-395. Given that this section is the main result (according to the aims of the analysis stated in the abstract and intro) I think it should go near the beginning of the results section. Unless I have misunderstood, the hypothesis tests in line 276 are part of the same result (is the variance component zero, if not how big is the variance component). So bringing these two sections together would improve the structure.

In the abstract in the results (word limit allowing), I think it would be useful to state the result that 55% of the variability in malaria outcomes could be attributed to the differences betwen SEAs. This to me is the most useful statement of the results.

Line 418: I am glad that you spend some time to discuss the surprising result that mosquito net use predicts increased malaria prevalence. However I think you have perhaps missed a likely cause of this estimate. I would guess that the positive relationship is because nets are delivered to areas with high malaria burden and people are more likely to use the nets if they feel malaria is a big risk to them. This argument does however raise the problem that the causality might be reversed here (malaria causes nets).

Line 197: You have written the sigma2b is an estimate of what *could* be explained. Quite subject but I would think of it as what *needed to be* explained. The random effect could explain everything if you removed everything.

Language issues

----------------

Abstract, methods. A sample of 9272 individuals were *used* in this

Abstract, conclusions. The methods used in this study have allowed *us* to make

Line 28: the survey design organizes the.

Line 29: *They* then select. Or something like that.

Line 36: or enumeration area, may have more similar.

Line 44: This model also does not allow us to study the association

Line 46: The intra-cluster correlation can *give* useful. Or something.

Line 54: regression models allow *analysts* to take into

Line 59: within-cluster.

Throughout, check whether within-cluster and between-cluster have hyphens.

Line 65: as a set of control variables, *the analysis* may provide information *on* how

Line 70: Also, it does not allow *analysts* to examine *the* influence of

Equation 2 and 3: I think the squared term in the denominators should come after the i|j. {w_{i|j}}^2 if you use latex.

Line 186: The *results from the* weighted analyses across

Line 187: and variance *parameters* of the

Line 188: results obtained *were* slightly

Line without a number after equation 5: identical for all the SEAs and bj *quantifies*

Line without a number beforeequation 6: The third model *was* defined

Just after line 233: The VPC measures *the* proportion

Line 315: to *have* had malaria than

Line 332: their interpretations *are* natural

Line 426: This *implies* that

Line 446: explanatory variables *were* introduced

Line 447: Hence, there is still a large

Line 456: have allowed *us* to make

Line 461: Such *an* approach yield*s*

Line 483: environment *are* interrelated, interpretation

Ref 29. The first author is Rabe‐Hesketh S I believe.

Ref 35 Type in neighbourhood.

6. PLOS authors have the option to publish the peer review history of their article (what does this mean?). If published, this will include your full peer review and any attached files.

Reviewer #1: No

---

## [Author Response · Author response to Decision Letter 0]

23 Oct 2021

Response to the Academic Editor and Reviewer # 1 comments on the manuscript:

“Multilevel logistic regression modelling to quantify variation in malaria

prevalence in Ethiopia” by Bereket Tessema Zewude, Legesse Kassa Debusho;

Tadele Akeba Diriba

Manuscript ID: PONE-D-21-12914

Dear Editor,

We are very grateful to the Academic Editor and Reviewer # 1 for the thorough review

of our manuscript and for their insightful and detailed comments. The marked-up copy

of our manuscript that highlights changes made to the original version has been

uploaded as a separate file labeled “Revised Manuscript with Track Changes”. We hope

we have addressed all the points that the Academic Editor and Reviewer # 1 have

made and provide the following summary of our changes.

Academic Editor Comments

1) Overall the English language is good but the manuscript needs a final copy-edit

before publication. E.g. 2.4 million (not millions) in the abstract. Line 36 should

read ‘may have *more* similar’. There are other small errors.

Authors’ response:

In the revised version of the manuscript, this and other errors that were identified by the

Academic Editor and Reviewer # 1 as well as by the authors have been corrected.

2) Introduction: Please update the malaria burden numbers from 2018 to more

recent data (e.g. using the latest World Malaria Report)

Authors’ response:

We have considered the Academic Editor recommendation and revised the malaria

burden numbers from 2018 to 2019 using the World malaria report 2020, please see

Lines 9-15 and 23-25 on page 2.

3) Table 1. Please briefly explain in the legend the difference between the 3

models. The table should be broadly understandable to the reader without

referring to the main text.

Authors’ response:

We have made a change in the legend to address the Academic Editor suggestion, that

is, the following statement was used as a legend:

“Estimated regression coefficients (SE) and variance components for the models with

cluster-specific random effects bj only (M0), with the individual / household-level

predictors and bj (M1), and M1 with cluster-specific predictors (M2).”

4) The abstract is rather hard to follow for a generalist reader. Please consider

rewriting this sentence: “The main objective of this study was to quantify the

variance component of cluster effects and cluster characteristics on individual

malaria rapid diagnostic tests (RDT) outcome.” Perhaps you could say

something like ‘to quantify the extent of clustering of malaria infection as

assessed by RDT’.

Similarly the results section of the abstract is hard to read. Can you name some

of the most important variables for explaining the clustering, which you found?

Authors’ response:

We have considered the Academic Editor recommendation and revised the Abstract.

The changes are marked by red on page 1 on the revised version of the manuscript.

5) I suggest moving the final paragraph which discusses malaria climate further up

in the discussion, since this is a key point. Please reference other studies which

have sought to explain malaria prevalence by both climate variables as well as

MIS variables. For example the Malaria Atlas Project is a major group which has

been working on this problem for many year. E.g. see the map tab on the Malaria

Atlas Project forecasts of malaria prevalence for Ethiopia

https://malariaatlas.org/trends/country/ETH What are the differences in the

variables used for analysis in this study or other similar geospatial models

compared to yours?

Authors’ response:

These comments also incorporated in the revised version. The final paragraph in the

“Conclusion” section of the old version was moved to the last part of the “Discussion”

section, please see Lines 457-511 on pages 14-15. In addition, based on the Academic

Editor’s comment, we have added some information related to the Malaria Atlas Project

(MAP), the limitations of our work, and what MAP provides. Then made

recommendation for further research to address the gaps.

6) I suggest removing the phrase ‘best alternative’ for analysis on line 398, since

there are many other sophisticated methods which seek to do this, e.g.

geospatial modelling as referenced above.

Authors’ response:

The phrase “best alternative” has been removed in the revised version, please see Line

406 on page 13.

Reviewer # 1 Comments

1) Minor points

My main problem with the manuscript in it's current state is that the policy

importance of the main aim of the analysis is not well explained and is not trivially

obvious.

What policy implications does a large (or small) between-cluster variance

component have?

I can imagine some.

Maybe it suggests that we are still missing covariates important for predicting

malaria.

If so, what might those covariates be and what are the barriers to collecting that

data?

Is it that nonlinear models and interaction terms are needed?

Or is it that there are cluster level drivers that we will never be able to measure?

What does this mean for policy, that we an never make excellent predictions of

malaria or that the only way to guide resource distribution is to measure malaria

prevalence in ever cluster in a country?

Authors’ response:

We have incorporated the following in Lines 474-492 on page 15 of the revised version:

“As discussed in the result section, the results of this study show that there is still high

variation between survey enumeration areas (SEAs) or clusters in malaria RDT

outcomes after taking into account the effects of individual / household characteristics

and the two cluster effects. The variation could be due to bio-ecological or human

factors or other covariates that are important for predicting malaria prevalence but not

included in the study. These factors may help to guide malaria control interventions and

improve their efficiency and effectiveness, and hence may lead to more effective public

health strategies and could have important policy implications for health promotion.” 

We hope this and the information that followed address the Reviewer # 1 concern.

2) Relatedly, while you cite some references that use random effects models, you

don't cite any estimates of between-cluster variance from any other study.

While I recognise that these estimates might be difficult to find in many papers, if

this is the main aim of the analysis, any previous attempts at estimating this

value should be mentioned, if nothing else to help us set our expectations.

Have previous estimates found that between-cluster variation is very high or very

low?

Authors’ response:

We have added the following literature review information in the “Introduction” section in

Lines 57-67 on page 3 to address the issues raised by Reviewer # 1:

“There are two multilevel studies in Ethiopia, Peterson et al. [15] included village as a

cluster and concluded that village-level effects were important and Woyessa et al. [16]

considered household as a cluster and investigated individuals and household effects

but their study was conducted within one city in Oromiya region. The studies in Peru

[17] and Cambodia [18] also included household as a cluster in their multilevel

analyses. The later showed the importance of household-level risk factors in explaining

differences in infection prevalence between households and between villages. Whereas

the former compared risk factors in community-specific models and reported malaria

prevalence differences between communities were influenced by individual and

household variables but variances of the random household-level effects were too

small.”

3) I'm not sure if this point is relevant but in some senses, the cross-validation error

in models of malaria prevalence are a measure of the same quantity.

It is the amount of malaria prevalence that cannot be predicted with the

covariates in the model.

Authors’ response:

The point has merit, but we think that it is not related to the objectives of the current

study.

4) The paragraph in lines 80-84 seems very unecessary to me. The structure of the

paper is totally standard.

Authors’ response:

We have taken out the above lines in the revised version.

5) Lines 243-245. I'm not sure what this line means. Are you saying that you

needed to seperately calculate the weights and tell PROC GLIMMIX to use

them? Or that you have provided the weights in an attached dataset? Or

something else.

Authors’ response:

We have rephrased the statement to make it clearer in revised version, please see

Lines 255-257 on page 9.

6) The section in lines 251-257 that explain how variable selection was performed

should be in the methods somewhere.

Authors’ response:

We have moved the variable selection information to “Methodology” section in the

revised version, please see Lines 204-209 on page 7.

7) Lines 367-395. Given that this section is the main result (according to the aims of

the analysis stated in the abstract and intro) I think it should go near the

beginning of the results section. Unless I have misunderstood, the hypothesis

tests in line 276 are part of the same result (is the variance component zero, if

not how big is the variance component). So bringing these two sections together

would improve the structure.

Authors’ response:

We have considered the Reviewer # 1 recommendation and we have moved the

“Quantify variation: Variance partition coefficient and Median odds ratio” section to Lines

335-364 on page 12. However, to keep the consistency of the results presentation, we

have not moved the hypothesis tests for variance components, or we have not joined

the two sections. The first subsection of “Results” section focuses on fitted multilevel

models where we have discussed the tests for model parameters including the variance

components.

8) In the abstract in the results (word limit allowing), I think it would be useful to

state the result that 55% of the variability in malaria outcomes could be attributed

to the differences between SEAs. This to me is the most useful statement of the

results.

Authors’ response:

We have considered the Reviewer # 1 recommendation and have amended the abstract

accordingly, please see the “Results” section of the abstract.

9) Line 418: I am glad that you spend some time to discuss the surprising result that

mosquito net use predicts increased malaria prevalence. However I think you

have perhaps missed a likely cause of this estimate. I would guess that the

positive relationship is because nets are delivered to areas with high malaria

burden and people are more likely to use the nets if they feel malaria is a big risk

to them. This argument does however raise the problem that the causality might

be reversed here (malaria causes nets).

Authors’ response:

We have found the phrase in red in the above Reviewer # 1 recommendation very

relevant and we have included it in the revised version, please see Lines 426-427 on

page 14.

10) Line 197: You have written the sigma2b is an estimate of what *could* be

explained. Quite subject but I would think of it as what *needed to be* explained.

The random effect could explain everything if you removed everything.

Authors’ response:

We have made a correction in Line 203 on page 7 as it has been suggested by

Reviewer # 1.

11) Language issues

We have also made all the language related corrections as suggested by Reviewer # 1.

Below we listed the line number and page where the suggested corrections have been

made.

• Abstract, methods. A sample of 9272 individuals were *used* in this

The correction has been made in Line 1 of “Methods” section of the abstract on

page 1.

• Abstract, conclusions. The methods used in this study have allowed *us* to make

The correction has been made in Line 1 of “Conclusions” section of the abstract

on page 2.

• Line 28: the survey design organizes the.

The correction has been made in Line 28 on page 2.

• Line 29: *They* then select. Or something like that.

The correction has been made in Line 29 on page 2.

• Line 36: or enumeration area, may have more similar.

The correction has been made in Line 36 on page 2.

• Line 44: This model also does not allow us to study the association

The correction has been made in Line 44 on page 3.

• Line 46: The intra-cluster correlation can *give* useful. Or something.

The correction has been made in Lines 46-47 on page 3.

• Line 54: regression models allow *analysts* to take into

The correction has been made in Line 54 on page 3.

• Line 59: within-cluster.

The correction has been made in Line 70 on page 3.

• Throughout, check whether within-cluster and between-cluster have hyphens.

We have made changes wherever necessary.

• Line 65: as a set of control variables, *the analysis* may provide information *on*

how

The corrections have been made in Lines 76 and 77 on page 3.

• Line 70: Also, it does not allow *analysts* to examine *the* influence of

The corrections have been made in Line 81 on page 3.

• Equation 2 and 3: I think the squared term in the denominators should come after

the i|j. {w_{i|j}}^2 if you use latex.

The corrections have been made in the two equations as it was suggested on

page 6.

• Line 186: The *results from the* weighted analyses across

The correction has been made in Line 192 on page 6.

• Line 187: and variance *parameters* of the

The correction has been made in Line 193 on page 6.

• Line 188: results obtained *were* slightly

The correction has been made in Line 194 on page 6.

• Line without a number after equation 5: identical for all the SEAs and bj

*quantifies*

The correction has been made in the second line from the bottom on page 6.

• Line without a number before equation 6: The third model *was* defined

The correction has been made in the third line after the equation on page 7.

• Just after line 233: The VPC measures *the* proportion

The correction has been made in the last line of page 8.

• Line 315: to *have* had malaria than

The correction has been made in Line 321 on page 10.

• Line 332: their interpretations *are* natural

The correction has been made in Line 369 on page 13.

• Line 426: This *implies* that

The correction has been made in Line 433 on page 14.

• Line 446: explanatory variables *were* introduced

The correction has been made in Line 453 on page 14.

• Line 447: Hence, there is still a large

The correction has been made in Line 454 on page 14.

• Line 456: have allowed *us* to make

The correction has been made in Line 518 on page 16.

• Line 461: Such *an* approach yield*s*

The corrections have been made in Line 523 on page 16.

• Line 483: environment *are* interrelated, interpretation

The correction has been made in Line 473 on page 15.

• Ref 29. The first author is Rabe-Hesketh S I believe.

The correction has been made in Reference 33 on page 18.

• Ref 35 Type in neighbourhood.

The correction has been made in Reference 39 on page 18.

---

## [Decision Letter · Decision Letter 1]

2 Feb 2022

PONE-D-21-12914R1Multilevel Logistic Regression Modelling to Quantify

Variation in Malaria Prevalence in EthiopiaPLOS ONE

Dear Dr. Zewude,

Thank you for submitting your manuscript to PLOS ONE. After careful consideration, we feel that it has merit but does not fully meet PLOS ONE’s publication criteria as it currently stands. Therefore, we invite you to submit a revised version of the manuscript that addresses the points raised during the review process.

There has been a reassignment of editor since the last version. The previous reviewer has kindly accepted to review again, noting still a concern with the policy interpretation. The new editor has checked the previous editorial assessment both for the previous version of this manuscript and the earlier version, while also looking in detail at the present manuscript. It is felt there is enough information to make a decision, although the editor has been unable to revise the supplementary material that should be provided in a more accessible format such as pdf.

There are currently a number of issues that have to be addressed. In a general manner, PLOS ONE strongly suggests the use of the STROBE or similar checklists to ensure that reporting is complete (https://www.strobe-statement.org/download/strobe-checklist-cross-sectional-studies-pdf). You can include in the revision the checklist specifying the line numbers where each aspect is addressed. I see things missing, starting with the rationale, the population of interest, who was tested for malaria, everybody in the household? A sample?

Rationale: The authors do not provide the rationale for studying MIS2011 when a more recent MIS2015, https://www.malariasurveys.org/documents/Ethiopia_MIS_2015.pdf , is available. They should discuss the rationale why this might be still interesting. They mention the survey as ref [6] but they do not say why they have not used MIS 2015. They should also explicitly mention the trends. They mention the trend from 2015-2019. What happened from 2011 to 2015? That should come first since this is precisely after this study.The literature revision does not include multilevel studies using MIS. They exist, should be mentioned and acknowledged and compared in the discussion. There is in particular a study using Ethiopia MIS 2015, https://doi.org/10.1186/s12889-020-09560-1 . See also https://doi.org/10.1016/j.ijid.2020.12.062 . Also the authors should not oversell the novelty of the study. Using multilevel modelling is standard in many fields and using sample weights with complex survey structure is a requirement in order to get results representative at the National level. It is not a plus to use them. It is just wrong if you do not use them here. It might be right if a different sampling strategy is used. Overall the introduction can be shortened considerablyThere is a need to edit the language. Please make a thorough language check. Example of sentences that can be improved: “The results also show that the individuals vulnerability to malaria infection decreases significantly with increase in age”. Maybe “increasing age” or “for older, maybe separate sentencesYour main result on cluster-level variance should refer to the best model. And that is the model with covariates. However the abstract states “About 55% of the variability in malaria RDT outcomes of individuals could be attributed to the difference between clusters. However, when accounting for the individual / household- and cluster-level characteristics, part of the variability which is relevant at the cluster-level became lower.”. Please provide only the figure for the best overall model according to AIC or BIC (say explicitly which criteria you are using) mentioning that the percentage variability is “after controlling for ….”. On the other hand, you might report M1 instead of M2 since M2 precisely aims at explaining what is behind the cluster variation (with no change in the individual level controls).Make clear in the abstract the population you are analyzing: Eg: “A sample of 9272 individuals” [representative of the National population? The adult population? This is never very clear]. Also: why are you focusing only on three states? How are these states compared to the National average?The referee is concerned with the lack of policy feedback gained from the multilevel analysis. While it is true that more recent data being available, this has less of an impact, the policy consequence of finding that a large proportion of variance remains at the cluster level is that policy should focus on those hotspots of malaria. In order for the analysis to be useful what would be interesting is to know where these malaria hotspots are located. You should provide a map showing the cluster random effects to guide action and comment on them (do they move along regional lines, are they very local, …).In the introduction you mention two levels of effects, the cluster and the household. However, the methods section only has random effects at the cluster level. This should be listed as a limitation since there are obvious correlation among malaria status at the household level. Why are you not including this? That would strengthen the paper since part of the currently described as cluster variance must be household-level variance and you can potentially separate one from the other?I have been unable to look at the supplementary material and it is fundamental to see whether PLOS ONE statistical guidelines are being followed since there are many missing elements in the main text. Please provide it in an edited format, eg. Pdf. Not tex.In table 1 provide fit statistics for model 0.The names of the models are different in table 1 and 2. Is model 2 model 1 and model 3 model 2?

A more comprehensive assessment needs to wait to see the supplementary material. Just make sure that all the issues raised in the strobe statement and PLOS guidelines for statistical reporting in (https://journals.plos.org/plosone/s/submission-guidelines).

We look forward to receiving your revised manuscript.

Kind regards,

José Antonio Ortega, Ph.D.

Academic Editor

PLOS ONE

Reviewers' comments:

Reviewer's Responses to Questions

**Comments to the Author**

1. If the authors have adequately addressed your comments raised in a previous round of review and you feel that this manuscript is now acceptable for publication, you may indicate that here to bypass the “Comments to the Author” section, enter your conflict of interest statement in the “Confidential to Editor” section, and submit your "Accept" recommendation.

Reviewer #1: (No Response)

2. Is the manuscript technically sound, and do the data support the conclusions?

Reviewer #1: Yes

3. Has the statistical analysis been performed appropriately and rigorously? 

Reviewer #1: Yes

4. Have the authors made all data underlying the findings in their manuscript fully available?

Reviewer #1: Yes

5. Is the manuscript presented in an intelligible fashion and written in standard English?

Reviewer #1: Yes

6. Review Comments to the Author

Reviewer #1: Rereeview for "Multilevel Logistic Regression Modelling to Quantify

Variation in Malaria Prevalence in Ethiopia"

PLOS one

PONE-D-21-12914

Review

---------

The authors have made a number of language edits in line with the editors comments and my own. These are all fine.

None of my comments in my original review were very important. However, the comment that I thought was most important related to setting the study into it's policy context. Some of the results, such as estimating which factors (age, temperature etc.) are related to high malaria, have obvious policy implications, but are also the same thing that people have been estimating for years. The big novelty in this study that I could see was the focus on estimating the between cluster variance after accounting for covariates (this is stated as such in line 456). However, the policy importance of the between cluster variance wasn't clear to me; how does a Ministry of Health use this information to guide malaria control for example?

In response the authors have added "The variation could be due to bio-ecological or human factors or other covariates that are important for predicting malaria prevalence but not included in the study. These factors may help to guide malaria control interventions and improve their efficiency and effectiveness, and hence may lead to more effective public health strategies and could have important policy implications for health promotion."

This states that the results are important for policy but still don't say how or why they are important. Again, what specific actions might a ministry take that they wouldn't have had taken without this information?

I'm afraid this isn't a particularly good or helpful rereview. I'm just saying the same thing I said last time. But I don't believe the authors have really fixed the issue.

7. PLOS authors have the option to publish the peer review history of their article (what does this mean?). If published, this will include your full peer review and any attached files.

Reviewer #1: No

---

## [Author Response · Author response to Decision Letter 1]

2 May 2022

Response to the Academic Editor and Reviewer # 1 comments on the manuscript:

“Multilevel logistic regression modelling to quantify variation in malaria

prevalence in Ethiopia” by Bereket Tessema Zewude, Legesse Kassa Debusho;

Tadele Akeba Diriba

Manuscript ID: PONE-D-21-12914R1

Dear Editor,

We are very grateful to the Academic Editor and Reviewer # 1 for the thorough review

of our manuscript and for their insightful and detailed comments. The marked-up copy

of our manuscript that highlights changes made to the original version has been

uploaded as a separate file labeled “Revised Manuscript with Track Changes”. We hope

we have addressed all the points that the Academic Editor and Reviewer # 1 have

made and provide the following summary of our changes.

Academic Editor Comments

I see things missing, starting with the rationale, the population of interest, who was

tested for malaria, everybody in the household? A sample?

Authors’ response:

In the revised version of the manuscript, we have included information that explains why

the three regions were considered, please see Lines 166-169 on page 4 Further, the

other part addressed in the “Response and explanatory variables” section, please see

Lines 173-177 On page 4.

1) Rationale: The authors do not provide the rationale for studying MIS2011 when a

more recent MIS2015,

https://www.malariasurveys.org/documents/Ethiopia_MIS_2015.pdf, is available.

They should discuss the rationale why this might be still interesting. They

mention the survey as ref [6] but they do not say why they have not used MIS

2015. They should also explicitly mention the trends. They mention the trend

from 2015-2019. What happened from 2011 to 2015? That should come first

since this is precisely after this study.

Authors’ response:

The authors have requested for 2015 malaria indicator survey data, the response that

we got was “the archived data didn't appear to be full and in the required format”. Then

EPHI has shared few datasets, which has no information on the survey design, such as

strata, cluster and sampling weights. Since this information vital for our research, we

could not consider the MIS2015 data that was provided to us. The authors assume

including this justification in the manuscript is not appropriate.2

We have considered the Academic Editor recommendation and revised the

“Introduction” section to address the trend issue, please see Lines 57-75 on page 2.

2) The literature revision does not include multilevel studies using MIS. They exist,

should be mentioned and acknowledged and compared in the discussion. There

is in particular a study using Ethiopia MIS 2015, https://doi.org/10.1186/s12889-

020-09560-1 . See also https://doi.org/10.1016/j.ijid.2020.12.062 . Also the

authors should not oversell the novelty of the study. Using multilevel modelling is

standard in many fields and using sample weights with complex survey structure

is a requirement in order to get results representative at the National level. It is

not a plus to use them. It is just wrong if you do not use them here. It might be

right if a different sampling strategy is used. Overall the introduction can be

shortened considerably

Authors’ response:

We have considered the Academic Editor recommendation and revised the

“Introduction” section to include the above suggested literatures and others, please see

Lines 115-119 on page 3.

However, please note that in the first article the authors have used the sampling weights

only to compute the descriptive statistics not for modelling purpose, whereas in the

second articles the authors fitted unweighted multivariate models. Otherwise, we fully

agree that “using sample weights with complex survey structure is a requirement in

order to get results representative at the National level”. As we have mentioned in the

“Introduction” section, however this is not the case in most of the existing literatures.

3) There is a need to edit the language. Please make a thorough language check.

Example of sentences that can be improved: “The results also show that the

individuals vulnerability to malaria infection decreases significantly with increase

in age”. Maybe “increasing age” or “for older, maybe separate sentences

Authors’ response:

In the revised version of the manuscript, this and other errors that were identified by the

authors have been corrected.

4) Your main result on cluster-level variance should refer to the best model. And

that is the model with covariates. However the abstract states “About 55% of the

variability in malaria RDT outcomes of individuals could be attributed to the

difference between clusters. However, when accounting for the individual /

household- and cluster-level characteristics, part of the variability which is

relevant at the cluster-level became lower.”. Please provide only the figure for the

best overall model according to AIC or BIC (say explicitly which criteria you are3

using) mentioning that the percentage variability is “after controlling for ….”. On

the other hand, you might report M1 instead of M2 since M2 precisely aims at

explaining what is behind the cluster variation (with no change in the individual

level controls).

Authors’ response:

These comments also incorporated in the revised version. The Abstract was corrected,

the results of the best model were presented in “Results” section (please see Lines 361-

430 on pages 10-13) and the “Discussion” section also focused on these results (please

see Lines 477-495 & 499-519 on pages 14-15). The fit statistics or model selection

criteria are presented in Table 1 and since M2 is the best model its results are

presented in Table 2 separately.

5) Make clear in the abstract the population you are analyzing: Eg: “A sample of

9272 individuals” [representative of the National population? The adult

population? This is never very clear]. Also: why are you focusing only on three

states? How are these states compared to the National average?

Authors’ response:

We have made changes in the “Methods” section of the abstract to address the above

comment and similar changes were also made in the “Study data” section, please see

Lines 24-30 on page 1.

6) The referee is concerned with the lack of policy feedback gained from the

multilevel analysis. While it is true that more recent data being available, this has

less of an impact, the policy consequence of finding that a large proportion of

variance remains at the cluster level is that policy should focus on those hotspots

of malaria. In order for the analysis to be useful what would be interesting is to

know where these malaria hotspots are located. You should provide a map

showing the cluster random effects to guide action and comment on them (do

they move along regional lines, are they very local, …).

Authors’ response:

Based on the above comments we have added some results from spatial clustering

analysis supported by a figure which contains hot-spots areas or districts. Please note

that due to the geographical coordinates of sample households we could not able to

produce a map of hotspots with regions or districts names in one plot, we don’t have a

shape file used for the MIS rather we presented it as point plot, please see Figure 1 on

page 13. Then we have made comments or recommendations by referring the hot-spots

and significant factors,4

7) In the introduction you mention two levels of effects, the cluster and the

household. However, the methods section only has random effects at the cluster

level. This should be listed as a limitation since there are obvious correlation

among malaria status at the household level. Why are you not including this?

That would strengthen the paper since part of the currently described as cluster

variance must be household-level variance and you can potentially separate one

from the other?

Authors’ response:

In the revised section, we have given explanation in the “Multilevel logistic regression

model with random effects” section why we did not include a random effect for a

household-level, please see Lines 229-235 on page 5. It reads as follows:

“In addition, there is correlation among malaria status of individuals living in the same

households. These introduce intra-class correlation, which is a measure of the degree

of similarity among malaria status of members of the same cluster, i.e. SEA or

household. However, in the dataset used in this study only one member per household

was tested for malaria from 47.12% households in the sample. This could introduce

large number of missing cases if a within household variation considered.”

8) I have been unable to look at the supplementary material and it is fundamental to

see whether PLOS ONE statistical guidelines are being followed since there are

many missing elements in the main text. Please provide it in an edited format, eg.

Pdf. Not tex.

Authors’ response:

The authors apologize for uploading the text version of the Supplementary material in

the previous submission, now we have provided in pdf format.

9) In table 1 provide fit statistics for model 0.

Authors’ response:

In the revised version, we have provided the fit statistics for all fitted models in Table 1,

please see page 10.

10) The names of the models are different in table 1 and 2. Is model 2 model 1 and

model 3 model 2?

Authors’ response:

We have made correction in the revised version, please see Tables 1 and 2 on pages

10 and 12, respectively.5

Reviewer # 1 Comments

The authors have made a number of language edits in line with the editors comments

and my own. These are all fine.

None of my comments in my original review were very important. However, the

comment that I thought was most important related to setting the study into it's policy

context. Some of the results, such as estimating which factors (age, temperature etc.)

are related to high malaria, have obvious policy implications, but are also the same

thing that people have been estimating for years. The big novelty in this study that I

could see was the focus on estimating the between cluster variance after accounting for

covariates (this is stated as such in line 456). However, the policy importance of the

between cluster variance wasn't clear to me; how does a Ministry of Health use

this information to guide malaria control for example?

In response the authors have added "The variation could be due to bio-ecological or

human factors or other covariates that are important for predicting malaria prevalence

but not included in the study. These factors may help to guide malaria control

interventions and improve their efficiency and effectiveness, and hence may lead to

more effective public health strategies and could have important policy implications for

health promotion."

This states that the results are important for policy but still don't say how or why

they are important. Again, what specific actions might a ministry take that they

wouldn't have had taken without this information?

Authors’ response:

The main purpose of estimating between cluster variance in the study was to assess

between cluster variation and if there is a variation what percent of it explained by the

covariates in the model. The above quoted response was to mention that the

unexplained heterogeneity after considering the individual / household and SEA

characteristics or covariates possibly due to those variables not included in the model.

Now in the revised version, we have recommended specific action related to significant

factors or covariates and results from spatial clustering analysis, specifically hotspots.

---

## [Decision Letter · Decision Letter 2]

24 May 2022

PONE-D-21-12914R2Multilevel Logistic Regression Modelling to Quantify

Variation in Malaria Prevalence in EthiopiaPLOS ONE

Dear Dr. Zewude,

Thank you for submitting your manuscript to PLOS ONE. After careful consideration, we feel that it has merit but does not fully meet PLOS ONE’s publication criteria as it currently stands. Therefore, we invite you to submit a revised version of the manuscript that addresses the points raised during the review process.

Some of the issues raised have been solved but there are still many issues pending revision in the abstract, the text, and the supplementary material. On a general remark, while PLOS ONE is a general interdisciplinary journal, this is an applied work not a methodological contribution. The substantive implications of the analysis need to be highlighted, and the goal cannot be just the application of a method (which is already standard in many areas).

Line numbers are missing in the manuscript with track changes. This makes it difficult to edit the manuscript. Please include them next time.Abstract: I noted the rationale was missing and it is still missing. The background section cannot start as it does with “The 2011 Ethiopian National Malaria Indicator Surveys (EMIS) data used in this study have a multilevel structure where individuals in selected households are nested within cluster and this may result in dependent data”. You should start stating the purpose of your study and why is it relevant. The purpose also cannot be to apportion the variance by itself. It must be to apportion the variance of some measure of interest. You have reformulated the introduction but not the abstract.You repeat twice “malaria rapid diagnostic test outcomes” in the background. Better use “prevalence of malaria”, or even in the first appearance “the prevalence of malariaYou declare twice in the abstract that you focus on three regions but you do not list them. Please do so. Note also the clumsy writing of the first mention on “Further, the study was also interested in assessing factors that affect the malaria rapid diagnostic test outcomes of household members in three major regions of Ethiopia.”. That cannot be “also” an interest. You are ONLY studying the prevalence of malaria in these 3 regions, not anywhere else. This is connected to the previous comment. The background section in the abstract has to be completely rewritten.Note in the introduction you are also stating (p.3) similar things to the abstract. Please change also the introduction stating clearly the specific rationale at the beginning as it was suggested in the previous revision.In the rest of the abstract you are not providing the concrete results or conclusions. Please state your MAIN results (maximum of 3 or 4) in a quantitative way.Typos still present Eg: Supplemantary material is called, several times, “Supplementery Material”The supplementary material needs to be thoroughly revised. It cannot be so verbose. What you are describing is the application of a backwards elimination algorithm. You do not need to describe each step in detail with all the estimates. Usually backwards elimination is carried out directly through some algorithmic procedure and you just need to report the variables you sequentially remove and their effect on the selection criteria (eg: AIC, BIC). It seems you are doing it by hand based on the p-value of a joint significance test. You could report the first step (the complete model), a table with the list of variables removed at each subsequent step together with the p-value of the type III test and the AIC or BIC after removing the variable, and the last step.There is a need to copy-edit the manuscript. One more instance: “These three regions were considered due to the data sharing police of Ethiopian Public Health Institute that since we are not part of EMIS project team we had a limited access to the survey data for our research.” I’d suggest “The choice of these three regions was driven by the data-sharing limitations imposed by the Ethiopian Public Health Institute, the owner of the data”The description of the target population and the sampling strategy needs to go in the data section, not in the response and explanatory variables section.The policy implication in p.14 is unwarranted, “Therefore, the regional government should frequently provide IRS to households in malaria hot-spots areas and the population at risk, and further educate people about its advantage” given that you do not find a significant effect.You are not commenting in the discussion the apparent contradictory result that those with mosquito nets are more likely to be infected. Why should that be so? Is this in line with the literature and with results from Ethiopia?The paper is missing a table one with the proportions and mean values of variables, possibly by region, possibly according to the value of the dependent variable. Its absence makes interpretation very hard. Eg: Are mosquito nets very prevalent and its absence is telling us something about local conditions? Is spraying common… Please include it and use to it to inform the interpretation of results and the discussion.Figure 1 has too small a type. It cannot be read. Please submit your revised manuscript by Jul 08 2022 11:59PM. If you will need more time than this to complete your revisions, please reply to this message or contact the journal office at plosone@plos.org. Please include the following items when submitting your revised manuscript:A rebuttal letter that responds to each point raised by the academic editor and reviewer(s). You should upload this letter as a separate file labeled 'Response to Reviewers'.A marked-up copy of your manuscript that highlights changes made to the original version. You should upload this as a separate file labeled 'Revised Manuscript with Track Changes'.An unmarked version of your revised paper without tracked changes. You should upload this as a separate file labeled 'Manuscript'.

We look forward to receiving your revised manuscript.

Kind regards,

José Antonio Ortega, Ph.D.

Academic Editor

PLOS ONE

Reviewers' comments:

Reviewer's Responses to Questions

**Comments to the Author**

1. If the authors have adequately addressed your comments raised in a previous round of review and you feel that this manuscript is now acceptable for publication, you may indicate that here to bypass the “Comments to the Author” section, enter your conflict of interest statement in the “Confidential to Editor” section, and submit your "Accept" recommendation.

Reviewer #1: All comments have been addressed

2. Is the manuscript technically sound, and do the data support the conclusions?

Reviewer #1: Yes

3. Has the statistical analysis been performed appropriately and rigorously? 

Reviewer #1: No

4. Have the authors made all data underlying the findings in their manuscript fully available?

Reviewer #1: No

5. Is the manuscript presented in an intelligible fashion and written in standard English?

Reviewer #1: Yes

6. Review Comments to the Author

Reviewer #1: (No Response)

7. PLOS authors have the option to publish the peer review history of their article (what does this mean?). If published, this will include your full peer review and any attached files.

Reviewer #1: **Yes: **Tim Lucas

---

## [Author Response · Author response to Decision Letter 2]

2 Aug 2022

Response to the Academic Editor comments on the manuscript:

“Multilevel logistic regression modelling to quantify variation in malaria

prevalence in Ethiopia” by Bereket Tessema Zewude, Legesse Kassa Debusho;

Tadele Akeba Diriba

Manuscript ID: PONE-D-21-12914R2 

Dear Editor,

We are very grateful to the Academic Editor for the thorough review of our revised manuscript and for the insightful and detailed comments. The marked-up copy of our manuscript that highlights changes made to the original version has been uploaded as a separate file labelled “Revised Manuscript with Track Changes”. We hope we have addressed all the points that the Academic Editor has made and provide the following summary of our changes.

Line numbers are missing in the manuscript with track changes. This makes it difficult to edit the manuscript. Please include them next time.

Authors’ response:

The authors apologize for uploading the revised manuscript without line numbers in

the previous submission. The authors have included the line numbers in the current revised version of the manuscript.

Abstract: I noted the rationale was missing and it is still missing. The background section cannot start as it does with “The 2011 Ethiopian National Malaria Indicator Surveys (EMIS) data used in this study have a multilevel structure where individuals in selected households are nested within cluster and this may result in dependent data”. You should start stating the purpose of your study and why is it relevant. The purpose also cannot be to apportion the variance by itself. It must be to apportion the variance of some measure of interest. You have reformulated the introduction but not the abstract.

Authors’ response:

We have considered the Academic Editor recommendation and revised the “Background” section of the Abstract to include the above comments, see page 1.

You repeat twice “malaria rapid diagnostic test outcomes” in the background. Better use “prevalence of malaria”, or even in the first appearance “the prevalence of malaria

Authors’ response:

We have considered the above comment and made changes in the “Background” section of the Abstract, see page 1.

You declare twice in the abstract that you focus on three regions but you do not list them. Please do so. Note also the clumsy writing of the first mention on “Further, the study was also interested in assessing factors that affect the malaria rapid diagnostic test outcomes of household members in three major regions of Ethiopia.”. That cannot be “also” an interest. You are ONLY studying the prevalence of malaria in these 3 regions, not anywhere else. This is connected to the previous comment. The background section in the abstract has to be completely rewritten.

Authors’ response:

We have rewritten the “Background” section in the Abstract, see page 1.

Note in the introduction you are also stating (p.3) similar things to the abstract. Please change also the introduction stating clearly the specific rationale at the beginning as it was suggested in the previous revision.

Authors’ response:

Based on the above comment, we have restructured the “Introduction” section, see pages 2-3.

In the rest of the abstract you are not providing the concrete results or conclusions. Please state your MAIN results (maximum of 3 or 4) in a quantitative way.

Authors’ response:

Based on the above comments we have rewritten the “Results” and “Conclusion” sections, see page 1.

Typos still present Eg: Supplemantary material is called, several times, “Supplementery Material”.

Authors’ response:

Based on the above comments we have replaced the phrase “Supplementery Material” by “Supplementary Material” in the current revised version of the manuscript and in the Supplementary material file wherever necessary. 

The supplementary material needs to be thoroughly revised. It cannot be so verbose. What you are describing is the application of a backwards elimination algorithm. You do not need to describe each step in detail with all the estimates. Usually backwards elimination is carried out directly through some algorithmic procedure and you just need to report the variables you sequentially remove and their effect on the selection criteria (eg: AIC, BIC). It seems you are doing it by hand based on the p-value of a joint significance test. You could report the first step (the complete model), a table with the list of variables removed at each subsequent step together with the p-value of the type III test and the AIC or BIC after removing the variable, and the last step.

Authors’ response:

The authors made changes in the current revised version of the manuscript “Supplementary Material” as it was suggested in the above comment.

There is a need to copy-edit the manuscript. One more instance: “These three regions were considered due to the data sharing police of Ethiopian Public Health Institute that since we are not part of EMIS project team we had a limited access to the survey data for our research.” I’d suggest “The choice of these three regions was driven by the data-sharing limitations imposed by the Ethiopian Public Health Institute, the owner of the data”.

Authors’ response:

In the current revised version of the manuscript, we have made changes in Lines 111-112 on page 3 as it was suggested above. 

 The description of the target population and the sampling strategy needs to go in the data section, not in the response and explanatory variables section.

Authors’ response:

The description of the target population and the sampling strategy was moved to the “Study data” section Lines 114-118 on page 4 in the current revised version of the manuscript.

 The policy implication in p.14 is unwarranted, “Therefore, the regional government should frequently provide IRS to households in malaria hot-spots areas and the population at risk, and further educate people about its advantage” given that you do not find a significant effect.

Authors’ response:

Due to the Academic Editor Comment stated in Item 13, in the current revised version of the manuscript contains additional information to strengthen the police implication that the authors stated in the previous submission. Please see results in Table 1 on page 11 and Lines 459-464 on page 17.

 You are not commenting in the discussion the apparent contradictory result that those with mosquito nets are more likely to be infected. Why should that be so? Is this in line with the literature and with results from Ethiopia?

Authors’ response:

In the current revised version of the manuscript, the authors have made comment, please see Lines 470-481 on page 17. It reads as follows:

“The current study showed that there was a positive association between household 

has mosquito nets and the prevalence of malaria in the study regions, this contradicts 

results reported in malaria literature. However, this association was statistically 

nonsignificant. The positive associate of having mosquito nets with the prevalence of 

malaria could be because mosquito nets are delivered to areas with high malaria burden or due to either households did not treat the nets with insecticide or inappropriate use of the nets or perhaps an individual or a household member was exposed to mosquito bites during other times of the day or evening when the net was not in use. This also could be due to household members did not get proper training on how to use the nets from local public health workers. In addition, less than half of the sampled households in the study regions had no mosquito nets, this may increase the vulnerability of individuals living in the households which had no mosquito nets to malaria infection.”

 The paper is missing a table one with the proportions and mean values of variables, possibly by region, possibly according to the value of the dependent variable. Its absence makes interpretation very hard. Eg: Are mosquito nets very prevalent and its absence is telling us something about local conditions? Is spraying common… Please include it and use to it to inform the interpretation of results and the discussion.

Authors’ response:

The table was provided on page 11 in the current revised version of the manuscript and the results in this table were discussed in Lines 260-295 on pages 9-10.

 Figure 1 has too small a type. It cannot be read.

Authors’ response:

The size of the fonts for Fig 1 (b) legend were increased

---

## [Editor Report · Decision Letter 3]

4 Aug 2022

Multilevel Logistic Regression Modelling to Quantify

Variation in Malaria Prevalence in Ethiopia

PONE-D-21-12914R3

Dear Dr. Zewude,

We’re pleased to inform you that your manuscript has been judged scientifically suitable for publication and will be formally accepted for publication once it meets all outstanding technical requirements.

Kind regards,

José Antonio Ortega, Ph.D.

Academic Editor

PLOS ONE

Additional Editor Comments (optional):

It is felt that the changes introduced have addressed the main concerns that prevented the paper from being acceptable for publication and the manuscript has gained in focus and relevance. The method section might still be too long, considering that the methods are standard, but it is OK to keep it as it is.
---

## [Editor Report · Acceptance letter]

10 Aug 2022

PONE-D-21-12914R3 

Multilevel Logistic Regression Modelling to Quantify Variation in Malaria Prevalence in Ethiopia 

Dear Dr. Zewude:

I'm pleased to inform you that your manuscript has been deemed suitable for publication in PLOS ONE. Congratulations! Your manuscript is now with our production department. 

Kind regards, 

on behalf of

Dr. José Antonio Ortega 

Academic Editor

PLOS ONE